# Behavior-dependent directional tuning in the human visual-navigation network

Matthias Nau [1,2 ✉], Tobias Navarro Schröder [1], Markus Frey [1,2] & Christian F. Doeller [1,2 ✉]

The brain derives cognitive maps from sensory experience that guide memory formation and behavior. Despite extensive efforts, it still remains unclear how the underlying population activity unfolds during spatial navigation and how it relates to memory performance. To examine these processes, we combined 7T-fMRI with a kernel-based encoding model of virtual navigation to map world-centered directional tuning across the human cortex. First, we present an in-depth analysis of directional tuning in visual, retrosplenial, parahippocampal and medial temporal cortices. Second, we show that tuning strength, width and topology of this directional code during memory-guided navigation depend on successful encoding of the environment. Finally, we show that participants' locomotory state influences this tuning in sensory and mnemonic regions such as the hippocampus. We demonstrate a direct link between neural population tuning and human cognition, where high-level memory processing interacts with network-wide visuospatial coding in the service of behavior.

[1] Kavli Institute for Systems Neuroscience, Centre for Neural Computation, The Egil and Pauline Braathen and Fred Kavli Centre for Cortical Microcircuits, NTNU, Trondheim, Norway. [2] Max Planck Institute for Human Cognitive and Brain Sciences, Leipzig, Germany. ✉email: matthias.nau@ntnu.no; doeller@cbs.mpg.de

Human scene-processing and navigation regions interface between the lower-level sensory and higher-level cognitive domain. They gradually construct world-centered mnemonic representations of the environment, a cognitive mapping process thought to culminate in the medial temporal lobe (MTL)[1–7]. Areas such as the retrosplenial[5] and the parahippocampal cortex represent the spatial layout[8] and 3D structure[9] of the currently viewed scene, as well as its relative openness[10] or boundaries[11]. Downstream regions like the entorhinal cortex and the hippocampus use this information to derive a stable representation of the world and one's own position, direction, and speed in it[12,13]. Together, such spatial representations are often referred to as cognitive map, and are thought to fundamentally shape our memories and guide behavior[14]. To understand this process, we believe that it is critical to study the neural population activity of these regions in a naturalistic setting and in light of the behavior they support.

A critical challenge the brain needs to solve to map the environment is keeping track of our own direction as we move. Previous studies revealed directional representations and activity related to heading perception in several areas, including the medial parietal lobe and retrosplenial cortex[15–18], the parahippocampal gyrus[17,19–23], the entorhinal/subicular cortex region[16,22–25], the thalamus[18], and the superior parietal cortex[26,27]. Most of these studies used dedicated and constrained directional judgment- and mental imagery tasks, and often examined direction in a self-centered frame of reference. To date, it remains unclear how cognitive mapping is mediated by the scene-processing and navigation network, and how active spatial behavior and memory relate to environmental processing in this pathway.

Here, we used 7T functional magnetic resonance imaging (fMRI) to monitor human brain activity during naturalistic virtual navigation in a spatial memory task (Fig. 1). Inspired by prior successes of encoding models in characterizing fMRI responses in other domains[28,29], we then developed an iterative kernel-based encoding model (Fig. 2) of the participants' navigation behavior (Supplementary Fig. 1A–C) to map directional tuning across the human cortex. In this framework, voxels are considered to be directionally tuned if their activity can be predicted based on world-centered virtual head direction (vHD), i.e., the direction a participant is facing within the virtual arena at each moment in time. We analyzed the impact of spatial memory performance and locomotory states on this tuning, focusing on scene-processing and navigation regions due to their proposed involvement in cognitive mapping[2–4]. These regions include the early visual and retrosplenial cortex, the parahippocampal gyrus, the entorhinal cortex, as well as the hippocampus.

Our objectives were twofold. First, we aimed to quantify and map directional tuning in the human scene-processing and navigation network during active spatial behavior. Second, we examined how this tuning relates to the participants' behavior and memory.

## Results

**Voxel-wise encoding modeling of virtual navigation behavior.** During fMRI scanning, participants freely navigated in a circular virtual reality (VR) arena via key presses while memorizing and reporting object locations within it (Fig. 1a). Across different trials, participants indicated the locations of these hidden objects by navigating to them. After each trial, they received feedback about the true object location before the next trial started. We then tracked the improvement in memory performance over trials by assessing the memory error, i.e., the Euclidean distance

**Fig. 1 Spatial memory task in virtual reality (VR). a** First-person and bird's-eye view of the VR environment in which participants navigated freely via key presses. The circular arena was surrounded by 12 landmarks matched in visual features (colored triangles). Across trials, participants memorized and reported object locations by navigating to them and pressing a "drop" button followed by feedback. **b** Object-location memory. Participants' memory performance improved as indicated by a decrease in memory error (Euclidean distance between drop and true location). The blue line and shaded area represent the mean and SEM of the memory error across participants. Data were smoothed with a moving average kernel of five trials. The inset on the bottom left depicts the median memory error across trials for single participants and as whisker–boxplot (center, median; box, 25th to 75th percentiles; whiskers, 1.5× interquartile range, $n = 20$ participants). Source data are provided as a Source Data file.

between true and remembered location in each trial (Fig. 1b; Supplementary Fig. 2A, B).

Our encoding model analysis comprises multiple individual steps. We modeled world-centered virtual head direction (vHD) using basis sets of circular–Gaussian vHD kernels (Fig. 2a). Next, we estimated voxel-wise weights for each kernel with ridge regression using a training data set (Fig. 2b; Supplementary Fig. 3A). We then used these weights to predict the time course of each voxel in an independent test set (Fig. 2c). We define directional tuning strength as the model performance, i.e., how well the model predicted the time course of a voxel or region. Critically, a positive tuning strength suggests that a voxel selectively represents some vHDs over others in a temporally stable manner. Finally, by iterating through multiple basis sets differing in the number and the full-width-at-half-maximum (FWHM) of the directional kernels (Supplementary Fig. 3B), we also estimated the corresponding tuning width of each voxel. Tuning width was defined as the FWHM of the kernels, leading to the best model performance. Finally, we examined the relationship between the tuning strength and width estimated for different regions, the participants' navigation behavior itself, as well as their performance in the spatial memory task.

Our results are presented in three sections. First, we establish how the vHD-encoding model works by mapping directional tuning strength and width across the cortex. Second, we

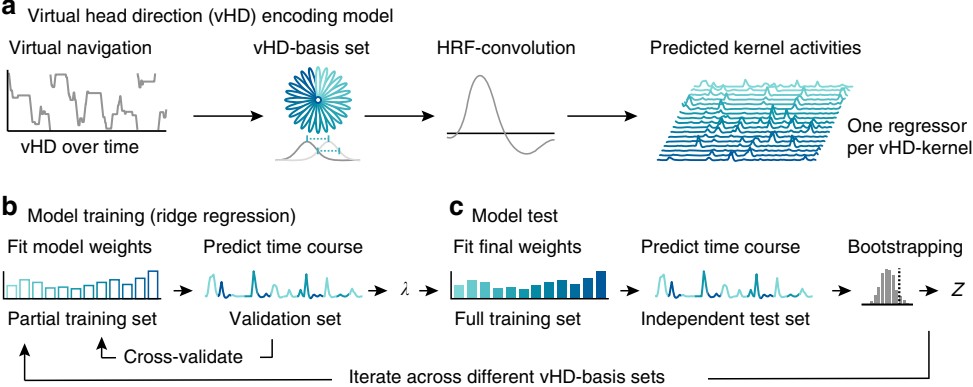

**Fig. 2 Analysis logic. a** Virtual head direction (vHD) encoding model. We modeled vHD using multiple basis sets of circular–Gaussian kernels covering the full 360°. Given the observed vHD, we generated predicted time courses (regressors) for all kernels in each basis set. The basis sets differed in the (full-width-at-half-maximum) kernel width and number. Spacing and width were always matched to avoid overrepresenting certain directions, i.e., the broader each individual kernel, the fewer kernels were used. The resulting regressors were convolved with the hemodynamic response function (SPM12) to link the kernel activity over time to the fMRI signal. **b** Model training. We estimated voxel-wise weights for each regressor in a training data set (80% of all data of a participant or four runs, 10 min each) using ridge regression. To estimate the L2-regularization parameter ($\lambda$), we again split the training set into partial training (60% data, three runs) and validation sets (20% data, one run). Weights were estimated in the partial training set, and then used to predict the time course of the validation set via Pearson correlation. This was repeated for ten values of $\lambda$ (log-spaced between 1 and 10.000.000) and cross-validated such that each training partition served as validation set once. We then used the $\lambda$ that resulted in the highest average Pearson's R to fit the final model weights using the full training set. **c** Model test: we used the final model weights to predict each voxel's time course in an independent test set (held-out 20% data, always the run halfway through the experiment) via Pearson correlation. These Pearson correlations were used to test model performance on a voxel-by-voxel level (Figs. 3 and 4). For a region-of-interest analysis (Figs. 5 and 6), we additionally converted model performance into Z scores via bootstrapping, ensuring that the results reflected the effects of kernel width and not of number. The null distribution of each voxel was obtained by weight shuffling ($k = 500$). Both model training and test were repeated for all basis sets.

demonstrate that the strength, width, and topology of this tuning depend on the participants' spatial memory performance. On a behavioral level, we show evidence that this likely relates to how well the environment has been encoded. Finally, we show that the tuning in both sensory and high-level mnemonic regions reflected the behavioral state of our participants, i.e., whether they were moving or not. Notably, our results cannot be explained by biases in sampling (Supplementary Fig. 1), model regularization, or data quality (Supplementary Fig. 3). Using simulated data, we further ensured that our analysis uncovers the true underlying tuning properties robustly across various noise levels and tuning profiles (unimodal, bimodal, and random) (Supplementary Fig. 4).

**Mapping directional tuning during spatial navigation.** Participants navigated in a VR environment, memorizing and reporting object locations within it. We used an iterative kernel-based voxel-wise encoding model of vHD to map directional tuning strength and width across the cortex (Fig. 2). The model performance is the Pearson correlation between the voxel time course predicted by the model and the one observed in the test set. The model prediction builds on weights estimated for each kernel using an independent model training procedure.

Our model successfully predicted activity in multiple regions in the ventral occipital, and medial parietal cortex as well as in the MTL (Fig. 3a). These regions overlap with known scene-processing and navigation regions, such as the retrosplenial cortex[5] and the parahippocampal cortex, as well as with the posterior hippocampal formation[3,4]. Along the parahippocampal long-axis, the tuning width followed a narrow-to-broad topology: narrow kernels best predicted activity in more posterior parts, wider kernels in anterior parts of the left-hemispheric parahippocampus (Fig. 3b). Visualizing the model weights showed that different voxels preferred different directions and that there were no distinct tuning profiles (uni-, bi-, and trimodal weight distributions) observable on the ROI level (Supplementary Fig. 5).

**Directional tuning reflects spatial memory performance.** After establishing that our vHD-encoding model did indeed predict activity in the visual-navigation network, we next asked whether the tuning was related to successful encoding of the environment and the object locations in it. We hypothesized that the tuning should be stronger in the ventral visual stream and MTL regions in participants that performed well in the spatial memory task. This hypothesis built on the idea that stronger tuning should indicate enhanced retrieval of directional information from high-level mnemonic systems.

To test this, we repeated the group-level analysis depicted in Fig. 3a, this time splitting the participants into two groups based on their across-trial median memory error (median split, Fig. 1b). We found that our model predicted activity in strikingly different networks in these two participant groups (Fig. 4a); the direction of these effects, however, was opposite of what we had predicted. In participants with low-memory error, i.e., good memory performance, the model predicted activity in the medial and ventral occipital lobe. Strikingly, in participants with high-memory error, the model predicted activity in the parahippocampal gyrus and in the MTL. A group-level permutation-based rank correlation between memory error and model performance further indicated that these differences build on a systematic relationship between directional tuning and spatial memory (Supplementary Fig. 6A–H). In both groups, we observed bilateral clusters in the medial parietal lobe. These clusters however barely overlapped between groups (Fig. 4b). In the low-memory-error group, the model predicted activity in a more anterior part, in the high-memory-error group in a more posterior part of the medial parietal lobe, akin to previous reports of an anterior–posterior functional distinction in this region[30].

To further characterize directional tuning explicitly in regions that derive world-centered representations of the visual environment, we next conducted a region-of-interest (ROI) analysis focusing on the early visual cortex (EVC), the retrosplenial

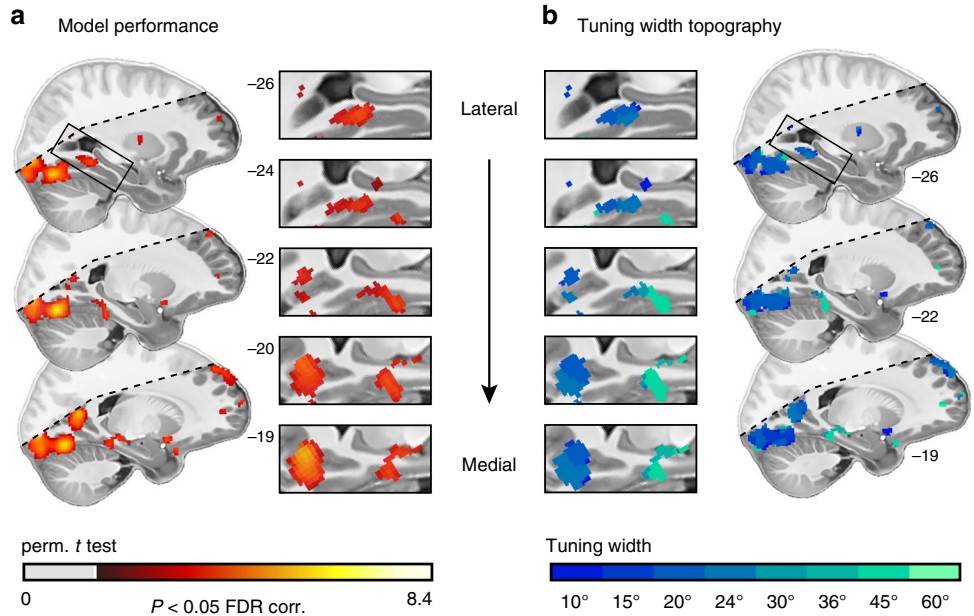

**Fig. 3 Mapping directional tuning across the human cortex (n = 20). a** Tuning map: directionally tuned voxels were determined by testing model performance against zero on group level using a permutation-based one-sample t test with k = 10,000 shuffles. We plot pseudo-T map thresholded at P < 0.05, FDR-corrected for all basis sets at T1 resolution overlaid on the group-average T1 scan. Approximate MNI coordinates were added. Inserts zoom in on the parahippocampal cortex and the medial parietal cortex. Multiple regions in the occipital lobe, the medial parietal and temporal lobes, and the parahippocampal gyrus were directionally tuned. **b** Tuning width: for each directionally tuned voxel, we color-coded the median tuning width (full-width-at-half-maximum of the directional kernels in the optimal basis set) that led to the highest pseudo-T value (depicted in **a**). The tuning width follows a narrow-to-broad topology along the parahippocampal long-axis. Shaded regions fell outside the scanning field of view in at least one participant. Source data are provided as a Source Data file.

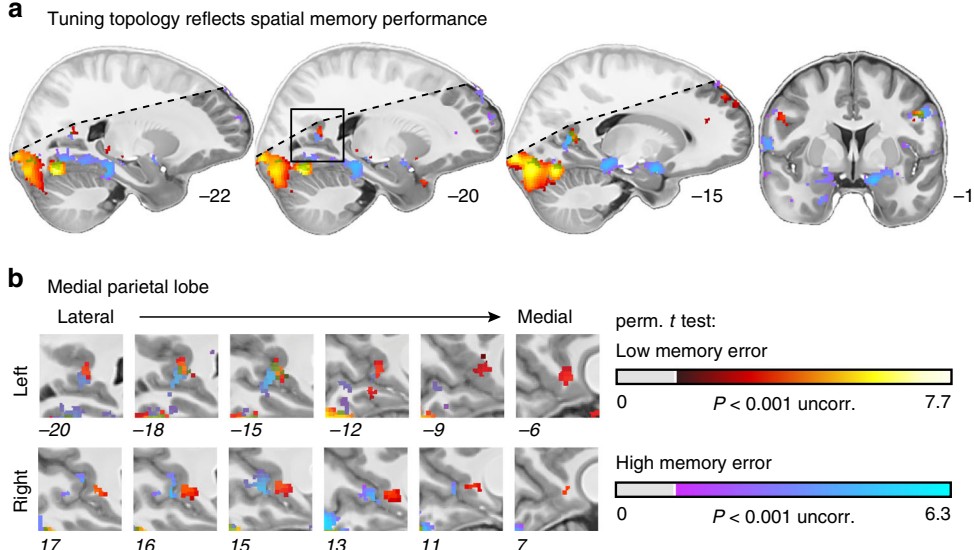

**Fig. 4 Directional tuning topology reflects spatial memory performance.** Participants were split into two groups depending on their across-trial median memory error (2 × n = 10). **a** Tuning map: directionally tuned voxels were determined by testing model performance against zero on group level using a permutation-based t test with 1024 unique random possible shuffles. This procedure results in a minimal possible P value of 0.00098, precluding FDR correction. We therefore plot pseudo-T maps thresholded at P < 0.001 uncorrected for all basis sets at T1 resolution overlaid on the group-average T1 scan. Hot colors depict results for the low-memory-error group, cool colors for the high-memory-error group. Approximate MNI coordinates added. **b** Zoomed-in depiction of the medial parietal lobe/retrosplenial cortex (RSC). There is an anterior–posterior distinction in directional tuning in RSC as a function of spatial memory performance. Shaded regions fell outside the scanning field of view in at least one participant. Source data are provided as a Source Data file.

cortex (RSC), parahippocampal gyrus (PHG), the posteromedial entorhinal cortex (pmEC), and the hippocampus (HPC) (Fig. 5a). We tested the pmEC subdivision of the entorhinal cortex because its rodent homolog region[31,32] is known to encode direction[33–35]. We obtained the vHD-model performance for every voxel in our

ROIs and every directional basis set as described before. For the ROI analysis, we added a bootstrapping procedure to ensure that our results reflected an effect of kernel width, and not of kernel number. For every basis set, we converted the Pearson correlations into Z scores via weight shuffling (Fig. 2c) and then

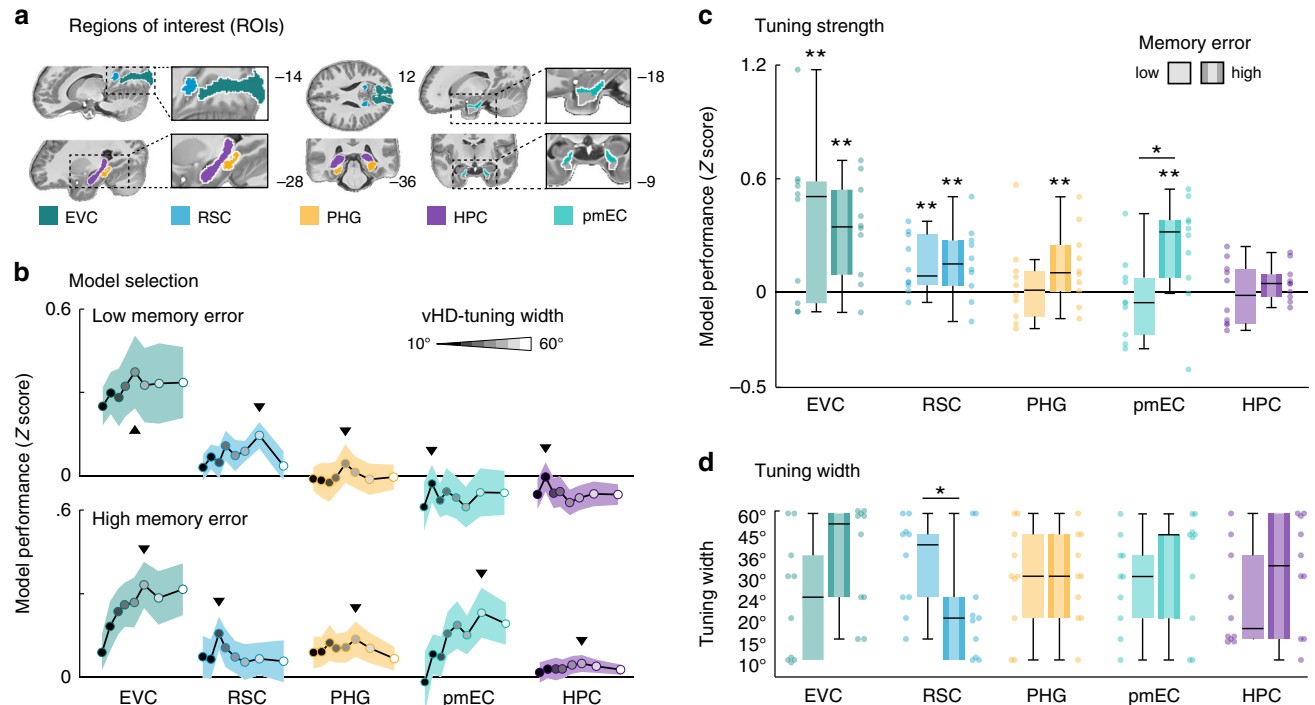

**Fig. 5 Region-of-interest (ROI) analysis. a** Human scene-processing and navigation regions tested in this study: early visual cortex (EVC), retrosplenial cortex (RSC), parahippocampal gyrus (PHG), posteromedial entorhinal cortex (pmEC), and the hippocampus (HPC). **b** Model selection: we plot the model performance (Z score) on ROI level for all basis sets. The black line and the shaded area represent the mean and SEM across participants. Each dot represents the group-average model performance for one basis set, with darker colors representing narrow kernels and lighter colors representing wider kernels. The following kernel widths were tested: 10°, 15°, 20°, 24°, 30°, 36°, 45°, and 60°. The black triangles mark the basis set that leads to the optimal model performance. **c** Optimal model performance for the two (high- and low-memory-error) participant groups. We plot single-participant data and group-level whisker–boxplots (center, median; box, 25th to 75th percentiles; whiskers, 1.5× interquartile range, $n = 2 \times 10$ participants). We observed directional tuning in EVC and RSC in both groups. In PHG and pmEC, this tuning depended on spatial memory performance. **d** Optimal tuning width. Similar to **b**, **c** we plot the tuning width that led to the optimal model performance selected on individual participant level. Participants with low-memory error had wider tuning than the ones with high-memory error. Hence, unlike tuning strength, the tuning width in RSC reflected spatial memory performance. One-sided (within-group) and two-sided (across-group) permutation-based t-test results were added: **$P < 0.05$, FDR-corrected, *$P < 0.05$, uncorrected. Source data are provided as a Source Data file.

averaged across the 25% most reliable training voxels in each ROI to reduce noise (see "Methods" for shuffling and voxel selection details). The resulting Z scores expressed how well the model predicted the activity relative to the voxel's null distribution. For each participant group and ROI, we selected the basis set that led to the best model performance on average (Fig. 5b). If a given region was not directionally tuned, the corresponding Z scores should be zero. We tested this on group level (one-tailed) as well as a difference between groups (two-tailed) using permutation-based t tests (Fig. 5c, see "Methods" for details). We observed that EVC and RSC encoded direction in both participant groups (EVC: low-memory error: $t(9) = 2.85$, $P = 0.014$, pFDR = 0.048, $d = 0.90$, CI = [0.67, 1.13]; high-memory error: $t(9) = 4.00$, $P = 0.004$, pFDR = 0.027, $d = 1.26$, CI = [0.99, 1.53]; RSC, low: $t(9) = 3.04$, $P = 0.006$, pFDR = 0.041, $d = 0.96$, CI = [0.69, 1.23]; high: $t(9) = 2.62$, $P = 0.015$, pFDR=0.040, $d = 0.83$, CI = [0.57, 1.08]). Importantly, in PHG and pmEC, such tuning was observed only in participants that had a high-memory error (PHG, low: $t(9) = 0.62$, $P = 0.320$, $d = 0.20$, CI = [−0.02, 0.41]; high: $t(9) = 2.16$, $P = 0.028$, pFDR=0.040, $d = 0.68$, CI = [0.43, 0.93]; pmEC, low: $t(9) = -0.42$, $P = 0.661$, $d = -0.13$, CI = [−0.40, 0.13]; high: $t(9) = 2.59$, $P = 0.020$, pFDR = 0.040, $d = 0.82$, CI = [0.54, 1.10]), in line with the voxel-wise group results (Fig. 4). In pmEC, the tuning strength additionally differed between groups ($t(18) = 2.32$, $P = 0.036$, $d = 1.04$, CI = [0.93, 1.15]). To again test whether this group difference reflected a

systematic relationship between tuning strength and memory error, we conducted a post hoc permutation-based rank correlation between memory error and pmEC model performance on the ROI level, which indeed seconded these results (rho = 0.48, $P = 0.035$, $k = 10,000$, Supplementary Fig. 6G). We did not observe such correlation in the EVC (rho = 0.11, $P = 0.636$, $k = 10,000$, Supplementary Fig. 6H). Notably, while the rodent homolog of pmEC is known to encode world-centered direction[33–35], another entorhinal subregion, the anterolateral entorhinal cortex (alEC), is not. Consistently, we observed directional tuning only in pmEC, not in alEC (Supplementary Fig. 7A–E).

In addition to the tuning strength, our approach also allowed to estimate the tuning width for each ROI. For each individual participant and ROI, we selected the tuning width that led to the optimal model performance (Supplementary Fig. 8) and compared it between groups. Strikingly, while the above-mentioned model performance in RSC was matched, the tuning width differed between groups ($t(18) = 2.04$, $P = 0.044$, $d = 0.91$, CI = [0.80, 1.02]). This suggests that participants with high-memory error might show a sharper tuning in RSC than participants with low-memory error (Fig. 5d).

In sum, while EVC, RSC, PHG, and pmEC were directionally tuned in at least one of the participant groups, tuning strength in pmEC as well as tuning width in RSC strikingly reflected how well participants performed in the spatial memory task. This is in line

with our hypothesis that the tuning should indicate whether the environment has been successfully encoded or not. However, we had hypothesized that stronger directional tuning in higher-level visual and MTL regions should be associated with better spatial memory performance. Our empirical test suggested the opposite. Why would mnemonic regions be more directionally tuned in participants that performed poorly in the spatial memory task? Contrary to our initial hypothesis, the differences between groups could reflect general differences in the cognitive strategy used or the participants' ongoing effort in encoding rather than retrieving a map of the environment. To investigate these effects post hoc in more detail, we examined how directional tuning strength developed in the course of the experiment. We performed leave-one-run-out cross-validation of our full modeling pipeline to obtain the directional tuning strength for each ROI not only for our original test run but for all runs. We found three main patterns of results (Supplementary Fig. 8C): EVC and RSC tuning strength did again not reflect memory performance, also not when cross-validated over runs. In contrast, PHG tuning strength did reflect memory performance robustly over runs. Strikingly, HPC and EC tuning also reflected memory performance, but the relationship between the two variables developed over time. Over scanning runs, tuning strength tended to increase in high-error participants and tended to decrease in low-error participants. Interestingly, on a behavioral level, we found that both groups approached the same level of memory performance in the course of the experiment (Supplementary Fig. 2A, B), but that the low-memory-error group had approached this performance level earlier in the experiment than the high-memory-error group (Supplementary Fig. 2B, C).

**Directional tuning reflects the behavioral state**. To investigate the relationship between directional tuning and the participants' behavior further, we repeated the above-described ROI analysis twice, once only modeling periods in which the participants moved, and once in which they stood still. In both cases, the participants rotated (Supplementary Fig. 1A, B). We compared these two scenarios by contrasting the respective model performances (Fig. 6a), revealing a positive effect of locomotion on tuning strength in EVC and RSC (EVC: locomotion: $t(19) = 4.04$, $P = 0.0004$, pFDR = 0.003, $d = 0.90$, CI = [0.79, 1.01]; stationary: $t(19) = 1.38$, $p = 0.091$, $d = 0.31$, CI = [0.20, 0.42]; contrast: $t(19) = 2.06$, $P = 0.049$, $d = 0.46$, CI = [0.35, 0.57]; RSC: locomotion: $t(19) = 3.68$, $P = 0.001$, pFDR = 0.004, $d = 0.82$, CI = [0.71, 0.93]; stationary: $t(19) = -0.82$, $P = 0.79$, $d = -0.18$, CI = [-0.29, -0.07]; contrast: $t(19) = 2.20$, $P = 0.043$, $d = 0.49$, CI = [0.38, 0.60]). Conversely, the activity in PHG and HPC could be better predicted while participants stood still (PHG: locomotion: $t(19) = -0.91$, $P = 0.799$, $d = -0.20$, CI = [-0.31, -0.10]; stationary: $t(19) = 2.37$, $P = 0.015$, $d = 0.53$, CI = [0.42, 0.64]; contrast: $t(19) = -2.35$, $P = 0.024$, $d = -0.52$, CI = [-0.63, -0.42]; HPC: locomotion: $t(19) = -3.48$, $P = 0.999$, $d = -0.78$, CI = [-0.89, -0.67]; stationary: $t(19) = 1.77$, $P = 0.046$, $d = 0.40$, CI = [0.29, 0.50]; contrast: $t(19) = -3.18$, $P = 0.004$, pFDR = 0.028, $d = -0.71$, CI = [-0.82, -0.60], Fig. 5c). Our results suggest that the directional tuning in human scene-processing and navigation regions reflects not only spatial memory performance, but also the behavioral state of the participants (i.e., whether they move or not). Notably, the fact that the differences in tuning strength and width were region-specific (Fig. 6a) suggests that our results cannot be explained by general differences in statistical power between the two behavioral states.

## Discussion
The present study investigated fMRI proxies of neural population activity reflecting directional coding during active spatial

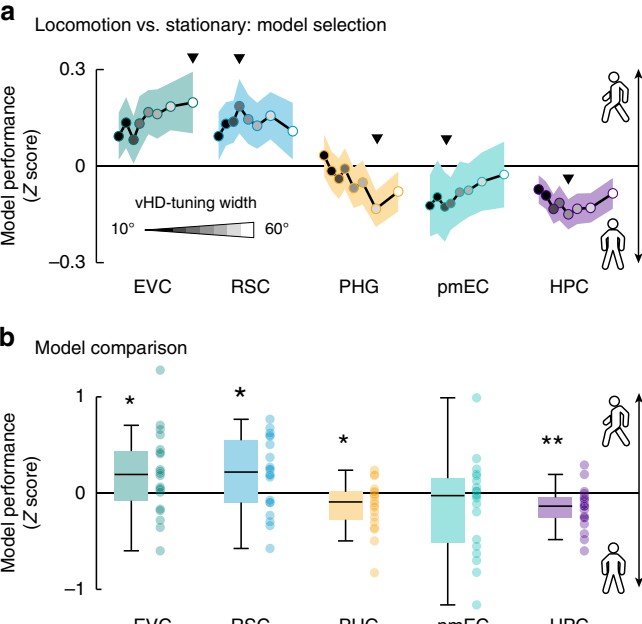

**Fig. 6 Behavioral-state analysis.** The analysis described in Fig. 2 was repeated, this time separating periods when participants navigated and when they stood still. **a** Model selection. We plot the difference in model performance (Z score) between locomotion and stationary periods across tuning widths (grayscale dots represent tuning width: narrow:dark, wide: light, also see Fig. 5). Positive values indicate that voxel time courses in an ROI could be better predicted when participants locomoted. Negative values indicate the opposite, with better model performance during stationary periods. Triangles mark the kernel width, leading to the strongest difference between models. **b** Model comparison. We plot the difference in model performance indicated in (**a**) as single-participant data and group-level whisker–boxplots (center, median; box, 25th to 75th percentiles; whiskers, 1.5× interquartile range, n = 20 participants). EVC and RSC tended to be better predicted during locomotion; PHG and HPC could be better predicted during stationary periods. These results suggest that the tuning in visual and mnemonic regions depends on the locomotory state. Two-sided permutation-based t-test results were added: **P < 0.05, FDR-corrected, *P < 0.05, uncorrected. Source data are provided as a Source Data file.

behavior. We put a focus on human scene-processing and navigation regions due to their proposed involvement in cognitive mapping: they derive world-centered mnemonic representations of the environment from sensory experiences. We used 7T-fMRI to monitor brain activity of participants navigating in a virtual environment and performing a spatial memory task. We developed an iterative kernel-based encoding model of the navigation behavior to map directional tuning across the human cortex. In addition, we examine its relationship to behavior and memory in detail. Visual, retrosplenial, parahippocampal, entorhinal, and hippocampal regions showed distinct response profiles, with a narrow-to-broad tuning width topology along the left-hemispheric parahippocampal long-axis. Furthermore, we examined the relationship between the tuning in each region, the participants' navigation behavior, and the performance in the spatial memory task. We found that the tuning in the RSC and pmEC, and likely in the parahippocampal gyrus, reflected how accurately participants reported the location of objects in the environment. Strikingly, while it was the tuning strength in pmEC and PHG, it was the tuning width and topology in RSC that depended on memory, notably however, with a large variability across participants. The strength, width, and topology of

directional tuning were therefore associated with the spatial memory performance of the participants, which demonstrates a direct link between neural population coding and cognition. The direction of this effect and how it developed over time in MTL regions speaks to the idea that the two participant groups might follow different cognitive strategies, and that the tuning reflects how well the environment has been encoded. Finally, the tuning in visual, retrosplenial, and parahippocampal regions, but especially in the hippocampus, additionally signaled the behavioral state of the participants.

Our observations emphasize the central role of scene-processing and navigation regions in spatial cognition[1–7,36], and are consistent with previous work on directional representations in the human brain[15–22,24,25]. They are also consistent with lesion studies showing that damage to regions like the RSC can impair the ability to orient oneself relative to landmarks[37]. Importantly, this study goes beyond previous reports in several aspects. Most studies used dedicated and constrained directional judgment- and imagery tasks to reveal directional representations in the brain. Here, we examined directional tuning during active naturalistic navigation (in VR) and additionally demonstrate that it depends on multiple behavioral factors. Many of these studies also examined self-centered directional coding, whereas our approach examines vHD explicitly in a world-centered frame of reference. Furthermore, using an iterative encoding model, we were also able to extract additional parameters such as the tuning width from fMRI responses. Similar approaches have been used for example to map the retinotopic[38] and semantic organization of the cortex[39,40], the 3D-depth tuning of scene-processing regions[9], to identify perceived stimuli from brain activity[41,42], and to reconstruct the online content of working memory[43,44]. Unlike previous work, the encoding model developed here (Fig. 2) does not build on information about a stimulus, but importantly is informed by the behavior of our participants directly.

An open question is whether the tuning observed here reflects processing of visual information or head direction (HD). Neurons representing world-centered HD, or HD cells, are abundant in the brain and have been studied most intensively not only in rodents[45], but also in monkeys[46]. HD cells reference facing direction relative to known landmarks[47] and are often compared with an internal compass mediating our sense of direction[45,48]. This HD-cell compass plays a central role in cognitive mapping and is thought to mediate homing, reorientation, and path-integration behavior[45,48–53]. We observed that multiple brain areas encoded direction, many of which overlap with regions known to contain HD cells in rodents and monkeys. These regions include the RSC[54–56], the postsubiculum (part of the hippocampal formation)[57], and the entorhinal cortex (EC)[33–35]. The latter consists of at least two subdivisions in rodents, the medial (MEC) and the lateral entorhinal cortex (LEC), likely corresponding to the pmEC and alEC in humans[31,32]. HD cells have been observed only in the MEC, not the LEC, paralleling our observations of world-centered directional tuning in the human pmEC, not in the alEC (Supplementary Fig. 7).

We also observed directional tuning in the early visual cortex for which no HD cells have been reported to date. This raises the question whether the effects reported here are due to HD, lower-level sensory, or landmark processing. We believe that these options are not exclusive, and the underlying processes might strongly interact in our naturalistic task. The only source of information to infer direction here were the landmarks, whose visibility at every moment depended on the position and the direction within the arena. Importantly, even if some landmarks were seen more often than others while walking in a certain direction, because they were all matched in low-level visual features, the only difference between directions was landmark

identity. Such landmark processing and the integration of landmarks across viewpoints has been suggested to be fundamental to anchor our sense of direction in space[58]. In addition, locomotion[59,60] and world-centered location[61,62] have been shown to modulate EVC activity in rodents. Here, we observed that locomotion had a positive effect on model performance in EVC and RSC, suggesting stronger directional tuning in these regions during running (Fig. 6). Also, in monkeys[63] and humans[64], the EVC can represent the velocity of motion in a world-centered frame of reference, likely due to feedback from higher-level areas. Critically, high-level mnemonic regions, such as the entorhinal cortex, are not known to be visually responsive, yet we still observed a directional code there. Consistent with this are reports of HD cells in MEC[33–35] as well as its well-connected position within the HD circuit[65]. Also, locomotion had a positive effect on model performance in EVC and a negative effect in the hippocampus, again contrasting higher-level MTL function against early visual processing. Areas such as the PHG code direction even in the absence of visual experience[19]. In addition, shared information and activity covariations between the EVC and the hippocampal formation suggest that early perceptual processing could well be modulated by higher-level cognitive processes[6,61,62,66–68].

Interestingly, the fact that the directional tuning we observed depends on behavior is at odds with the activity profile of "classical" HD cells. First, these cells provide a directional representation that is continuous (i.e., they always maintain similar activity on population level following attractor dynamics)[45,53]. Second, impairments or even lesions of the HD-cell system only have very moderate effects on behavior[53]. However, growing evidence suggests that there are at least two types of HD cells in the brain: "classical" continuous HD cells, as well as non-continuous sensory HD cells. The latter have been shown to switch between active and inactive states, providing a directional representation that is controlled by visual landmarks instead of attractor dynamics[34]. Our present results are in line with the function and the location of these sensory HD cells found in the entorhinal and parahippocampal cortex[34], as well as potentially in the RSC[55]. Some of these cells also alternate between several preferred directions depending on context, potentially explaining the variation in RSC tuning width we observed (Fig. 5d). On a population level, directional activity referenced to multiple landmarks would likely be interpreted as having broader tuning curves than the stable unidirectional counterparts. While classical HD cells integrate vestibular inputs[45], which were not available in our VR task, visual information alone is sufficient to drive HD-cell-like coding in VR[69] and to determine its alignment to the environment[70]. Behavioral evidence further suggests that the visual environment, specifically its geometry, influences our sense of direction[71,72]. In our task, the VR arena was circular, and all landmarks were carefully matched with respect to their visual appearance (Fig. 1). In one half of the arena, the triangle-shaped landmarks faced upward, in the other half downward (Fig. 1b). The hypothetical axis in-between did not contain any landmarks and did not bias vHD sampling (Supplementary Fig. 1).

Taken together, low-level sensory information alone can explain neither the tuning we observed, nor its relationship to behavioral performance. Instead, our results are well in line with previous work on higher-level spatial representations and with landmark processing in the visual system. We did not disentangle visual from nonvisual factors here, but the tuning in the EVC and its progression over time clearly differed from the one in MTL regions. The relationship between directional tuning and spatial memory hence arose on a higher level of the cortical hierarchy, not the visual input stage. However, our results support the notion that these higher-level cognitive processes are closely

intertwined with visual processing up to its earliest stages in the brain.

Could viewing behavior have influenced the results? One limitation of the current study is that we cannot differentiate between vHD and the direction of gaze. Other studies using eye tracking indeed suggested that the landmarks capture most of the viewing[73]. Neurons that encode the allocentric location of gaze could hence in theory lead to a directionality on population level if location and gaze are correlated. Such cells exist in the primate hippocampus and encode allocentric view independent from head direction[74] or together with HD and self-location in a conjunctive code[75]. Computational models proposed that it is the conjunction between location and direction that mediates the environmental anchoring of HD in the RSC[76]. Gaze direction typically varies more broadly over time than vHD, potentially explaining the relatively broad tuning in EVC (Fig. 5), as well as the influence of locomotion on it (Fig. 6). If correct, this predicted that early visual cortex should show a sharper tuning when analyzed as a function of gaze direction compared with vHD. Future studies using eye tracking could address these questions directly. Acknowledging this ambiguity, we here refer to directional tuning more generally.

Why does directional tuning reflect spatial memory performance? We hypothesized that participants with stronger directional tuning in higher-level visual and mnemonic regions should perform better in the spatial memory task. We based this prediction on the idea that stronger tuning indicated enhanced retrieval of directional information from memory. Our empirical test showed the opposite: participants with stronger tuning in higher-level mnemonic regions performed worse in the spatial memory task. Neither data quality or model parameters (Supplementary Fig. 3), nor directional sampling (Supplementary Fig. 1) could explain these results. While the underlying mechanism remains unclear, we can speculate about its nature. For that, we feel it is important to revisit what tuning means in this context. The tuning strength describes how well the time course of a voxel could be predicted. This does not necessarily constitute a net increase or decrease in activity, but instead how well we could predict the fluctuations over time. A voxel that weakly but consistently follows a directional modulation could hence show stronger tuning than the one that strongly codes direction at very few time points.

Hypothetically, because fMRI likely measures synaptic processing rather than output spiking[77], the tuning could reflect how much inputs a region receives. For example, the MTL of high-error participants might receive stronger inputs from perceptual regions, because these participants were still in the process of forming a cognitive map, which heavily relies on visual information. Accordingly, low-error participants might have formed a cognitive map already. Consistent with this idea, we found that both participant groups approached the same level of memory performance, but that participants with stronger tuning in higher-level regions showed steeper learning curves than those with weaker tuning (Supplementary Fig. 2C). This is also consistent with earlier reports showing that hippocampal activity tracks the amount of knowledge obtained at a given time rather than the accumulated absolute knowledge[78], and with decreases in hippocampal activity in the course of spatial learning[79]. Analyzing how the tuning developed over time revealed that such continuous learning-related disengagement could however explain only the results observed for the low-, not of the high-memory-error group. The latter even showed slight increases in directional tuning over time in the entorhinal cortex (Supplementary Fig. 8C). One account that would reconcile these findings is that the memory-dependent effects could reflect a difference in cognitive strategy used, which might develop and get refined in the course of spatial learning.

Our observations are also in line with previous work on the emergence of spatial representations in scene-processing and navigation regions. For example, the RSC and PHG were shown to rapidly encode a novel environment by integrating information across different viewpoints and landmarks. Spatial representations emerging in the RSC were associated with a participant's ability to identify multiple scenes as belonging to the same or different location[80], as well as with wayfinding ability by registering landmark permanence[81]. Consistent with the present results, the RSC activity could hence signal whether an environment has been successfully encoded or not. We observed a broader tuning in RSC in well-compared to poorly performing participants, potentially indicating that the RSC processed information with a broader field of view. This could aid the integration of different landmarks and viewpoints[80,81]. Importantly, the medial parietal lobe, which comprises the RSC, is known for both its perceptual and mnemonic capacities[82], which might be topologically distributed. Previous work revealed an anterior–posterior distinction for scene construction and perception[3,30]. Our results (Fig. 4b) are consistent with this observation in two aspects. First, we also observed an anterior–posterior split on group level in bilateral medial parietal lobe as a function of memory performance (Fig. 4b). Second, the participant group with more anterior tuning also performed better in the spatial memory task. Given the above-mentioned reports, we speculate that this topology could indicate enhanced scene construction in participants with good memory performance. Again, this would also be consistent with the idea that these participants have already successfully encoded the environment and the object locations in it.

Notably, not only the RSC (Fig. 5c), but also the pmEC tuning (Fig. 5b) depended on the participants behavior, suggesting a different but related effect in pmEC. In line with this, rodent reorientation behavior depends on the stability of entorhinal HD-cell firing relative to the environment[52]. This alignment is also tightly coupled with the one of other spatial codes, such as the hippocampal place field map[83], which in turn predicts goal-oriented navigation behavior[84]. Since the entorhinal cortex is the key mediator of hippocampal–cortical communication[65], the present effects could therefore also reflect a related mechanism orchestrated by the hippocampus, with activity decreasing as environments become more familiar[85]. Finally, the effects could also reflect individual preferences for egocentric versus allocentric navigation strategies[86].

Looking forward, we see a wide range of future applications and exciting avenues to be explored. Future studies employing more dynamic model tests, such as inverted encoding modeling (proof of principle in Supplementary Fig. 9A–C), could reconstruct the modeled behavioral features more dynamically[28,29,87,88]. This could especially be helpful when combined with high temporal resolution measures, such as magnetoencephalography (MEG) or intracranial recordings, in combination with VR tasks[89] to monitor trial-by-trial changes in tuning as a function of behavioral performance. In addition, the influence of different sensory and behavioral variables on the present results could be further examined by removing the landmarks and testing which areas maintain their tuning, or by comparing the results to the ones of other visual encoding models.

Using virtual reality, a novel behavioral encoding model, and 7T-fMRI, we examined world-centered directional tuning during active spatial behavior in humans. We demonstrated such tuning in visual, retrosplenial, parahippocampal, and entorhinal cortices as well as the hippocampus. By mapping the tuning width across the cortex, we revealed a narrow-to-broad organization along the parahippocampal long-axis. Entorhinal and parahippocampal

tuning strength as well as retrosplenial tuning width and topology reflected how well participants performed in a spatial memory task. We provide evidence that these effects likely depend on the encoding of the environment and the object locations within it. Finally, we show that the tuning in visual, retrosplenial, and parahippocampal cortices as well as the hippocampus reflects the locomotory state. These results show the efficacy of encoding models for studying neural population dynamics during naturalistic navigation in human fMRI. They demonstrate a direct link between neural population coding and cognition, and show that high-level cognitive processes modulate directional tuning in the service of behavior.

## Methods

**Participants**. We recruited 26 participants for this study (11 females, 19–36 years old). Four participants were excluded because of excessive head motion, i.e., the number of instantaneous movements larger than 0.5 mm exceeded the across-participant average for more than one standard deviation. Another 2 participants were excluded because they finished fewer than four scanning runs. A total of 20 participants entered the analysis. The study was approved by the local research ethics committees (ethics committee University Duisburg-Essen, Germany and CMO region Arnhem–Nijmegen, NL) and participants gave written consent prior to scanning.

**Virtual reality task**. Participants performed a self-paced object-location memory task in virtual reality (Fig. 1a) adapted from Doeller and colleagues[22]. The circular virtual arena was created using the UnrealEngine2 Runtime software and was surrounded by 12 distinct landmarks positioned in steps of 30° and matched in visual similarity (triangles either tilted up- or downward, with red-, green-, and blue- colored corners). Participants could freely navigate in this arena via key presses. The smallest instantaneous rotational movement possible was 10° and translational movement speed was constant after a 500-ms ramp. In the beginning of the experiment, six everyday objects had to be collected, which were scattered across the arena. The location of these objects differed across participants, and the six objects were randomly drawn from a set of overall twelve objects, leading to unique location–object associations for each participant. Across different trials and without the objects being present, participants were prompted to navigate to the location of a previously cued object. After indicating the remembered location via key press (drop), the respective object appeared at the correct location to give feedback, and the participant collected the object again before the next trial began. After an average of 3 trials (range 2–4), a fixation cross was presented on a gray background for 4 s. An average of 179 trials were performed (range: 94–253 trials due to the self-paced nature of the task, Supplementary Fig. 2d) and object locations were randomized across participants. In order to explain the task and to familiarize participants with it, they performed a similar task on a desktop computer setup with different objects in a different virtual environment prior to scanning. We tracked the improvement in memory performance over trials by assessing the memory error, i.e., the Euclidean distance between true and remembered location in each trial measured in virtual vertices (arbitrary units).

**Behavioral analysis**. To ensure that there were no prominent or distinct directional cues that biased navigation behavior, we ruled out differences in the time spent facing in different directions. Supplementary Fig. 1 depicts the results of these analyses for all participants and time points, split into locomotion and stationary periods, as well as high- and low-memory-error participant groups. We accounted for individual differences in absolute time spent in the experiment by expressing time spent as percent of the total experimental duration. We binned vHD in steps of 10° and performed a repeated-measure (rm) ANOVA across directions, which did not reveal any biases in directional sampling ($F(35, 665) = 0.77$, $P = 0.834$), also not when testing the high-memory-error group ($F(35, 315) = 0.55$, $P = 0.984$) or the low-memory-error group ($F(35, 315) = 0.87$, $P = 0.675$) individually. In addition, we did not observe biases in directional sampling across the experiment when splitting the data into locomotion ($F(35, 665) = 0.79$, $P = 0.806$) and stationary ($F(35, 665) = 0.87$, $P = 0.681$) periods (Supplementary Fig. 1A).

In addition to the directional sampling in the course of the experiment, we analyzed the distribution of vHD within each TR. We again binned vHD into 10° steps and converted it into percent of total viewing time. We then circular-shifted each of the resulting histograms such that the most sampled direction lined up across TRs (Φ), revealing that participants spent 52% of the time within each TR facing into a single direction (Fig. 1b; Supplementary Fig. 1A). The distribution of vHD within each TR was therefore nonuniform and centered on one predominant direction. We used a two-tailed permutation-based unpaired t test to compare the time spent facing toward this predominant direction within each TR across across groups, which did not reveal a difference ($t(18) = −1.26$, $P = 0.224$, $d = −0.56$, CI $= [−0.68, −0.45]$, $k = 10,000$, Supplementary Fig. 1C). Using a paired version of this test, we did observe a difference between stationary and locomotion periods

($t(19) = 6.12$, $P = 0.0001$, $d = 2.74$, CI $= [2.63, 2.84]$, $k = 10,000$). Importantly, this had a differential effect for different regions of interests (Fig. 6), suggesting that there was no general positive or negative effect on model performance. During this task, participants spent around 54% of their time navigating, with shorter time spent rotating during locomotion compared with stationary periods (two-tailed permutation-based paired t test: $t(19) = 7.95$, $P = 0.0001$, $d = 3.55$, CI $= [3.44, 3.66]$, $k = 10,000$). There were no differences in the time spent translating ($t(18) = 0.41$, $P = 0.690$, $d = 0.18$, CI $= [0.07, 0.30]$, $k = 10,000$) or rotating between participant groups (during locomotion: $t(18) = 0.55$, $P = 0.625$, $d = 0.25$, CI $= [0.14, 0.35]$, $k = 10,000$ and during stationary periods: $t(18) = 1.00$, $P = 0.338$, $d = 0.45$, CI $= [0.33, 0.56]$, $k = 10,000$).

**MRI acquisition**. During the object-location memory task in VR, we acquired T2*-weighted functional images on a 7T Siemens MAGNETOM scanner using a 3D-EPI pulse sequence, a 32-channel head coil, and the following parameters: TR = 2756 ms, TE = 20 ms, flip angle = 14°, voxel size = 0.9 mm × 0.9 mm, slice thickness = 0.92 mm, slice oversampling = 8.3%, 96 slices with a 210-mm × 210-mm field of view, phase-encoding acceleration factor = 4, and 3D acceleration factor = 2. The first five volumes of each run were discarded. Functional images were acquired across 5 scanning runs of 210 TRs or approximately 10 min each. In addition, we acquired T1-weighted structural images (MP2RAGE, voxel size: 0.63-mm isotropic) and a B0-field map (gradient echo, voxel size: 1.8 × 1.8 × 2.2 mm) for each participant.

**Preprocessing**. The data used here were used in two previous reports[32,73]. Data were preprocessed using the automatic analysis library (https://github.com/automaticanalysis/automaticanalysis), utilizing functions of several analysis packages. For each participant, functional images were realigned and unwarped using SPM8, followed by independent component analysis (ICA) denoising using FIX artifact removal implemented in FSL 5.0.4. To improve the signal-to-noise ratio, and with it the ICA detection of noise components, data were smoothed with a Gaussian full-width-at-half-maximum kernel of 2.5 mm. Images were then nonlinearly normalized to a group-average EPI template using the Advanced Neuroimaging Toolbox (http://stnava.github.io/ANTs) and high-pass filtered with a 128-s cutoff using FSL. Voxel-wise variance explained by the six realignment parameters (x,y,z, pitch, roll, yaw) as well as by spikes (sudden deviations in signal intensity of more than two temporal standard deviations) was removed via nuisance regression. Out-of-brain voxels were excluded.

**Regions of interests (ROIs)**. In an ROI analysis, we tested human scene-processing and navigation regions (Fig. 5a) that were previously proposed to support cognitive mapping[1]. The hippocampal (HPC), anterolateral entorhinal (alEC), and posteromedial entorhinal (pmEC) ROIs were defined manually using ItK-SNAP (www.itksnap.org) based on the high-resolution group-average EPI template. The entorhinal masks were based on previous reports[32], in which the entorhinal mask was divided into anterolateral (alEC) and posteromedial entorhinal cortex (pmEC). The ROIs for the parahippocampal gyrus (PHG) as well as the retrosplenial cortex (RSC) were based on the reverse-inference meta-analysis for "Retrosplenial cortex" and "Parahippocampal cortex" using Neurosynth (https://neurosynth.org). We took the top 5% highest-probability voxels from each respective Neurosynth map and removed isolated voxels from the resulting binary masks. This procedure resulted in coherent bilateral clusters in the medial parietal cortex and parahippocampal gyrus, respectively. The early visual cortex (EVC) ROI was created by thresholding the corresponding probability map "Visual_hOc1" of the SPM anatomy toolbox at 50% and co-registering it nonlinearly to our group-average template space. To do so, we used SPM to first segment the group template and then to normalize the ROI into our template space using the resulting tissue maps and nearest-neighbor interpolation. The resulting ROI masks were located at the following average MNI coordinates [X, Y, Z] and were of the following size. EVC (gray matter only): left hemisphere [−4, −88, 0], right hemisphere [14, −86, 0], n voxels 2893 (median across participants), RSC: [−14, −56, 12] and [18, −54, 14], n voxels 1926, PHG: [−26, −38, −12] and [26, −34, −16], n voxels 2392, HPC: [−24, −24, −14] and [28, −22, −14], n voxels 9781, alEC: [−20, 0, −34] and [22, 0, −34], n voxels: 1693, pmEC: [−20, −10, −28] and [20, −8, −28], and n voxels: 1692.

**Analysis overview**. Our analysis estimated the directional tuning of a voxel in several steps. First, we built a vHD-encoding model by incorporating the participant's navigation behavior into basis sets of circular–Gaussian von-Mises distributions, which we call vHD kernels. Each individual direction was modeled with a different vHD kernel, each representing a smooth directional tuning without discretizing the data into bins. Second, we estimated voxel-wise weights for each of these kernels, together representing a voxel's tuning curve. We refer to this step as model training. Third, we used these weights to predict activity in held-out data that constituted the model test. This way, we obtained a measure of model performance for the given vHD-basis set. Finally, by iteratively varying the full-width-at-half-maximum of the vHD kernels in the basis set and repeating the above-mentioned steps, we not only tested one vHD-basis set, but multiple ones. This approach allowed us to also estimate the tuning width of each voxel (the kernel

width that maximized prediction accuracy). All processing steps mentioned comprised several individual substeps, which are described in detail below.

**Building the virtual head direction (vHD) encoding model.** We modeled vHD using a basis set of circular–Gaussian von-Mises distributions as implemented in the Circular Statistics Toolbox for Matlab (https://github.com/circstat/circstat-matlab). Each kernel in this basis set covered the full 360° with an angular resolution of 1°. Across different iterations of our analysis, we varied the full-width-at-half-maximum of these kernels, each time testing how well the resulting model weights allowed to predict activity in held-out data. To balance directional sensitivity across iterations, the spacing between kernels always matched the kernel width (i.e., the broader each individual kernel, the fewer kernels were used). We tested the following kernel widths, all representing divisibles of 360°: 10°, 15°, 20°, 24°, 30°, 36°, 45°, and 60°. Note that our region-of-interest analysis builds on weight shuffling to rule out that the number of kernels influenced model accuracies. For each kernel and participant, we computed the predicted kernel activities based on the vHD over time. To give an example, for a given vHD of 30°, a kernel centered on 30° was assigned high activity, a kernel centered on 40° slightly lower activity, and a kernel centered on 200° was assigned very low activity. By doing this for all time points and kernels, we built regressors representing the predicted activity, given a directional tuning and the specific vHD over time. Since vHD was sampled at higher temporal resolution than the imaging data, we then computed the within-TR activity of each kernel as the median activity across all time points within the TR. Finally, the resulting regressors were scaled from 0 to 1 and convolved with the hemodynamic response function (HRF) as implemented in SPM12 (https://www.fil.ion.ucl.ac.uk/spm/software/spm12/) using default settings (kernel length: 32 s, time to peak: 6 s). Each regressor represented a predicted activity profile over time as modeled by the respective kernel. To follow the example above, the activity of a voxel encoding the direction 30° should be more similar to the activity predicted by the kernel centered on 30°, than to the one predicted by the kernel centered on 200°.

**Model training.** We estimated voxel-wise weights for all vHD kernels in the corresponding basis set using l2-regularized (ridge) regression. To improve the directionality of this model, we added a covariate modeling movement independent of direction whose weight was discarded. Because vHD is not independent at two successive time points, the resulting design was multicollinear. Ridge regression avoids potential biases in the resulting weights by penalizing high coefficients, which could otherwise affect model accuracies. Since the regularization parameter ($\lambda$) cannot be known a priori, our model training builds on leave-one-out cross-validation to find the optimal $\lambda$ and with it the optimal model weights. As training set, we used the first two and the last two scanning runs, leaving the third run as the final and independent test set. A scan run typically took around 10 min. Since run 3 was acquired in the middle of the experiment, our results are invariant to the duration of the scanning session. If a participant did not complete all five runs, we always used the third valid run as a test set and all others as the training set. Within the training set, we used all runs, except one to fit voxel-wise weights for ten different regularization parameters log-spaced between 1 and 10,000,000, each time testing how well these weights predicted the activity in the left-out run. To assess prediction performance, we used Pearson correlation between the real time course of a voxel in the validation set and the time course predicted by vHD weighted by the estimated model weights. We cross-validated this prediction such that each run within the training set served as validation set once. The regularization parameter that led to the best prediction performance on average was then determined for each voxel. If no clear best-performing $\lambda$ could be determined (i.e., Pearson's R negatively approximated zero with increasing regularization), the respective voxel was excluded. Overall, this excluded ~15.5% of voxels (median across participants and model iterations), however only from determining the final $\lambda$, not from the full training-test procedure in which the final $\lambda$ was applied to all voxels. We then averaged $\lambda$ across voxels within each participant and used it to estimate the final model weights using the full training set. These model weights serve as the basis for all further model tests described below.

**Model test.** All model tests were performed on the held-out and independent test set (one scan run of around 10 min in the middle of the experiment). First, we predicted voxel-wise activity (similar to the model training) in a univariate forward-model approach. For each voxel, we generated a predicted time course by weighting the design matrix of the test run after HRF convolution by the model weights obtained for this voxel during model training. The resulting predicted time course was then compared with the observed time course using Pearson correlation. Note that all steps described below were repeated for multiple vHD-basis sets differing in the number and width of the corresponding vHD kernels.

We mapped directional tuning across the cortex using the statistical nonparametric mapping (SnPM) toolbox (http://warwick.ac.uk/snpm). We performed a permutation-based one-sample $t$ test of model performance (Pearson correlations) against zero ($k = 10,000$ shuffles, input image, and variance smoothing: 7.2 mm). To reduce computational costs, the preprocessed data were downsampled from 0.9 mm to 1.8 mm isotropic for this step. This was repeated for each vHD-basis set. We then used the SnPM toolbox to threshold the resulting

pseudo-T maps at a FDR-corrected $P < 0.05$. For each voxel, we then selected the across-participant median tuning width of the vHD-basis set that maximized the pseudo-T. Figure 3 depicts these results for all participants. For visualization, we plot the results overlaid on the group-average T1 template at T1 resolution obtained via nearest-neighbor interpolation. We repeated this analysis split into high-memory-error and low-memory error participants (i.e., median split of memory error, Fig. 4). Note that each participant group comprised n = 10 participants, resulting in 1024 possible shuffles and a minimal possible $P = 0.000977$. Because this precludes FDR correction, we use uncorrected $P = 0.001$ to visualize these subgroup effects.

In addition to the voxel-wise group analysis, we also conducted a region-of-interest analysis for areas involved in scene processing and navigation, such as the early visual cortex, retrosplenial cortex, parahippocampal cortex, the posteromedial entorhinal cortex, and the hippocampus (Fig. 5a). We again performed model training and test, this time focusing specifically on voxels in our ROIs. This greatly reduced the number of voxels and hence computational cost. To avoid potential influences of the number of kernels on these ROI results, we instead performed voxel-wise bootstrapping to convert the Pearson correlations into Z scores. The necessary null distribution of each voxel was obtained by shuffling the training weights 500 times, each time computing Pearson's R between the predicted and the observed time course of the test set. All shuffles were unique. Since not all voxels in our (probabilistic) ROIs were expected to carry vHD information and to increase robustness of our effects, we performed voxel selection within each ROI, limiting the model test to voxels with high predictability in the model training (top 25% highest prediction accuracy in the training). If a voxel was not directionally tuned in the training, we did not expect it to be directionally tuned in the test. Finally, the Z scores of the remaining voxels were averaged within each ROI. Again, we iterated this analysis for all basis sets (Fig. 5b), yielding the tuning strength (model performance) and the tuning width (the width of the vHD kernels in the basis set) of each ROI in the test set. While other model selection procedures are possible, ours enabled to estimate the tuning properties of a given region in a given run even if the tuning underwent changes across runs. Statistical inference was performed for each ROI and participant group (high- and low-memory-error participants) using permutation-based one-sample $t$ tests on group level (all possible 1024 shuffles, $n = 2 \times 10$) as implemented in the mult_comp_perm_t1 function distributed by Mathworks (https://se.mathworks.com/matlabcentral/fileexchange).

To test for differences across participant groups, we used permutation-based unpaired two-sample two-tailed $t$ tests ($k = 10,000$) as implemented in statcond distributed via the EEGLab Matlab toolbox (https://github.com/openroc/eeglab/blob/master/tags/EEGLAB7_0_0_0beta). For each permutation test, we report Cohen's d, including the bootstrapped confidence intervals as effect size. To examine how directional tuning developed over time, we further performed leave-one-run-out cross-validation across all scanning runs (Supplementary Fig. 8). We performed the ROI-level encoding modeling analysis (Fig. 5) for all runs, each time estimating the model weights on all runs but the one used for testing.

In addition to the forward-model test, we also inverted the encoding model to reconstruct, or decode, vHD multivariertly from the population of voxels within each ROI from our main test run 3 (Supplementary Fig. 9). We multiplied the Moore–Penrose pseudoinverse of all voxel-wise weights in an ROI ($m$ voxels $\times k$ weights)$^{-1}$ with the multivoxel pattern at each image acquisition (m voxels) to obtain the estimated vHD-kernel activities at each for volume (k weights). By doing this for all volumes, we reconstructed the vHD-kernel activities for the entire test set of each participant. To assess reconstruction performance, we used 2D correlation between the reconstructed kernel activities (k weights $\times$ n TRs) and the design matrix of the test run (also k weights $\times$ n TRs). Since again the number of kernels and hence reconstructed weights differed across iterations and basis sets, we used weight shuffling to convert the resulting correlations into Z scores. As in the forward model, we shuffled the model weights (500 random unique shuffles), each time going through the full vHD-reconstruction procedure. As expected, the results obtained by inverting the encoding model (Supplementary Fig. 9) resemble the ones obtained by the voxel-wise forward-model procedure (Fig. 5).

**Temporal signal-to-noise ratio (tSNR) and model regularization do not explain the results.** The estimated optimal regularization parameter $\lambda$ (Supplementary Fig. 3A) depended on the basis set (Supplementary Fig. 3B) (rmANOVA results: $F(7, 126) = 11.80$, $P = 1.7 \times 10^{-11}$) as expected, but not on participant group ($F(1, 18) = 0.0037$, $P = 0.952$) and there was no interaction between the two ($F(7,126) = 0.37$, $P = 0.920$). There were differences in tSNR across ROIs (rmA-NOVA results: $F(5, 90) = 258.29$, $P = 9.1 \times 10^{-52}$), but not across participant groups ($F(1, 18) = 0.082$, $P = 0.777$) and there was no interaction between the two ($F(5, 90) = 1.34$, $P = 0.255$) (Supplementary Fig. 3C). Neither model performance (Spearman correlation: rho = 0.025, $P = 0.788$), nor tuning width (rho = $-0.050$, $P = 0.587$) correlated with tSNR (Supplementary Fig. 3D).

**The model uncovers directional tuning robustly in simulated time courses.** To test the robustness of our model and to explore its limitations, we repeated the full model training and testing procedure for simulated voxel time courses with known tuning properties. As described in detail below, these simulations demonstrated that our model uncovers the correct kernel size robustly across various noise levels and tuning profiles (unimodal, bimodal, and random directional tuning).

First, we simulated voxel time courses with known tuning properties similar to how we built the design matrix for the actual modeling pipeline. To do so, we built the vHD kernels reflecting the desired tuning profile and then computed the corresponding time course based on the actually observed vHD of a randomly chosen sample participant. This has several key advantages over simulating completely random time courses, because it captures the natural dependencies and correlations between different directions and how these are sampled over time. Note, however, that the choice of participant does not change these results presented here. For each voxel, we simulated all five scanning runs.

We built simulated time courses (Supplementary Fig. 4A) for the assumed encoding of one random direction (unimodal model), two random directions (bimodal model), and randomly many random directions (random model) for each of the eight tuning width levels tested. The random numbers were sampled from a uniform distribution. Next, we added various degrees of Gaussian noise to these time courses ranging between one and ten standard deviations of the signal time course. For each of these noise levels and tuning widths, we simulated a total of 2500 voxels, approximating a typical ROI size in our data (see "Methods"). This resulted in 2500 unimodally tuned voxels for each of the 10 noise levels and each of the 8 different tuning widths for five simulated scanning runs; another 2500 bimodally tuned voxels and another 2500 randomly tuned voxels with equal noise, width levels, and runs.

Next, we ran the full encoding model pipeline as described in the paper on these simulated time courses and assessed model performance (i.e., the Pearson correlation between the observed and predicted (simulated) time course). We predicted to see that the kernel width that served as the basis for the simulated time course would also lead to the best model performance when tested. To test this, we averaged the model performance across voxels like in the actual fMRI analysis.

We observed that, unsurprisingly, increasing noise had a detrimental effect on model performance (Supplementary Fig. 4B). However, because our framework solely relies on the comparison between different kernel sizes within each noise level, we normalized the effect of noise on model performance for visualization. Interestingly, kernels smaller than the true kernel tended to lead to better model performance than kernels larger than the true kernel (Supplementary Fig. 4C). This likely arises from the fact that larger kernels can be approximated by smaller, but not easily by even larger ones. With increasing noise, the benefit of the smaller kernels vanished, possibly related to stronger overfitting to the noise. Note that in our imaging data, there was also no correlation between tSNR and model performance (Supplementary Fig. 3D).

Most importantly, and despite the partial benefits of smaller over larger kernels, the true kernel won over both smaller and larger kernels in all cases, independent of noise level or assumed tuning profile. As predicted, the true kernel width therefore always resulted in the best model prediction, at least when averaged across voxels (Supplementary Fig. 4C). In our main analyses, this averaging is done across voxels of an ROI, or by smoothing for the whole-brain group analyses (not the same as averaging but having a similar effect). In sum, the above-described simulations demonstrate that our behavioral encoding model framework uncovers the true underlying fMRI tuning robustly even in the presence of noise and independent of the actual tuning profile (unimodal, bimodal, or random).

**Reporting summary**. Further information on research design is available in the Nature Research Reporting Summary linked to this article.

## Data availability

The source data underlying Fig. 1b, 3a, b, 4a, b, 6b and Supplementary Figs. 1a–c, 2a–d, 3a, c, d, 4a–c, 5a, b, 6a–h, 7c–e, 8a–c and 9c are provided as a Source Data file. The virtual navigation data of a sample participant are provided together with analysis code (see "Code availability"). Other data are available from the authors upon reasonable request. Source data are provided with this paper.

## Code availability

We make available online our model simulation code in MATLAB that includes creating, fitting, and testing the encoding model on simulated time courses (https://osf.io/j5q9u/). This code can be easily adapted for new analyses and makes use of the virtual navigation data of a sample participant. Source data are provided with this paper.

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

## Acknowledgements
We thank Raphael Kaplan for helpful comments on an earlier version of this paper and Joshua B. Julian for helpful discussions. C.F.D.'s research is supported by the Max Planck Society, the Kavli Foundation, the European Research Council (ERC-CoG GEOCOG 724836), the Centre of Excellence scheme of the Research Council of Norway—Centre for Neural Computation (223262/F50), The Egil and Pauline Braathen and Fred Kavli Centre for Cortical Microcircuits, and the National Infrastructure scheme of the Research Council of Norway—NORBRAIN (197467/F50).

## Author contributions

M.N., T.N.S., and C.F.D. conceived the experiment. T.N.S. acquired and preprocessed the data. M.N. analyzed the data and wrote the paper with input from M.F., T.N.S., and C.F.D. All authors discussed the results and contributed to the paper.

## Competing interests

The authors declare no competing interests.
