## [Peer Review File · Nature Communications]

Reviewers' Comments:

Reviewer #1:

Remarks to the Author:

Summary

In this manuscript the authors lay out a very interesting and novel approach to the problem of quantifying heading sensitivity in human fMRI data. In short, they develop an encoding model for heading direction and demonstrate that it explains variance in the responses of a wide array of visual areas during a navigation task. They then present the parameters of this model to help unpack the nature of heading representations within and the potential particular contributions of various cortical regions. The manuscript is well written, the question timely and interesting, the computational approach novel to this domain, and the presentation of the data laudably both clear and transparent. There are however a number of concerns that must be addressed.

Relationship between Brain and Behavior

By far the most problematic observation in this paper is that better behavioral performance is associated with more prevalent directional tuning in the EVC and worse performance with tuning in the medial parietal and temporal cortices. This result just doesn't make sense, and highlights potential problems with the behavioral task. The task is framed as requiring heading information, but heading here is completely confounded with the presence of particular landmarks. These landmarks are consistent, visible, and change size with distance from them. Could one explanation be, regardless of the laudable attempt to make their features similar, that participants are coding particular landmarks or their cooccurrence in the visual field? This confound would explain the apparent sensitivity to EVC to heading, and the differences between the results during stationary/turning and locomotion periods via differences in the changes in the landmarks appearance (see Supp 1). While it is possible that head direction encoding and low-level sensory processing are not mutually exclusive, the inability to separate these two processes is clearly a weakness of the study. How are the difference between the high and low memory error groups to be interpreted, even within the medial temporal lobe.

An interesting experiment might be to remove the landmarks or even the entire arena during the participants response and see what of the apparent direction selectivity survives. Unfortunately, given the close spacing of the landmarks here, it seems unlikely that there is sufficient data when the landmarks were not visible or indicative of heading to analyze.

Tuning Width Analysis

The analysis of tuning width seems to suffer from a small circularity. While the basic fitting of the model is well validated in independent data, the tuning width is simply taken as the model that produced the best fits. Even though each of the models that is input into this selection is fair, selecting the best of the eight still introduces a significant bias. Here is some matlab code that illustrates the problem in random data that has no difference between the conditions:

```
randSet = rand(100,8,10000) >= .25;
allResult = squeeze(mean(randSet));
bestResult = squeeze(max(allResult));
avgResult = squeeze(mean(allResult));
[mean(avgResult) mean(bestResult)]
figure; histogram2(bestResult, avgResult); axis([.65 1 .65 1 0 Inf]); xlabel('Best'); ylabel('Avg');
```

To be clear, this problem likely does not detract significantly from many of the author's arguments but should be removed wherever possible, though this is sometimes tricky. For example, in Figure 3A, the depicted T-values are from the best model, but should be from the average, the currently presented

values are inflated. In Figure 3B, the tuning width presented is the average from independent circular selections of tuning widths, making the average likely okay to present, but no test is run to show that the T-values for the presented average widths are actually better than the other widths. In Figure 4C, significance markings are plotted against zero for the model performance, but these z-scores are artificially inflated and no such test should be performed. In Figure 4D, the variability amongst the participants in the tuning width metric is apparent, which makes some of the claims a bit difficult to rely on heavily. In general, I recommend that the authors carefully evaluate how best to present and analyze this metric and attenuate their claims about it.

Learning Slope Analysis

I am somewhat confused by the purpose of this analysis. The high-memory error group has room to improve and the low-memory error group does not. Why is it surprising that the slopes differ? Is there some specific point to this quantification?

Small Comments

1. The presentation of the memory error as essentially arbitrary units is unnecessarily opaque. These can be converted to proportions of the total arena diameter to give a better sense of the extent of the mislocalizations and the improvements over trials.
2. I couldn't find an explicit statement of what the various significance markings (*, **) corresponded to in the figure legends.

Reviewer #2:

Remarks to the Author:

In this manuscript, Nau and colleagues investigate directional tuning in the human brain using 7T-fMRI and a virtual navigation task. Building on previous successes of encoding models in the orientation selectivity domain, the authors find directional tuning across several regions of the ventral occipital, medial temporal, and medial parietal lobes. They also report that voxel time courses in mnemonic regions could be better predicted when subjects performed poorly (!) on a spatial memory task, and that tuning in several regions depended on whether subjects were moving.

The manuscript is technically strong, extremely clear, and very well written. My main critique is that, having established the effectiveness of this encoding model approach, the study misses on the opportunity to deliver more novelty such as delving deeper on the links to behavior. To be clear, I believe the manuscript is very strong and there are no major flaws, so the comments below are only a few suggestions of how I believe it could be made stronger.

MAJOR COMMENTS:

1. The authors hypothesize that "tuning should be stronger in ventral visual stream and medial temporal regions in participants that performed well in the spatial memory task". Instead, they found that tuning was stronger in those regions in participants with high memory error. They speculate that the observed group differences could be explained by the fact that learning takes longer in participants with high memory error. I think that this is an interesting hypothesis and could be explored further. For instance, could the authors test whether directional tuning reduces over time (e.g. looking at the correlation between predicted and observed time courses either on a voxel-by-voxel or ROI analysis)? If it does, does it reduce more quickly in participants with low memory error?
2. What is the tuning profile of a "typical" voxel (if there's such a thing as a typical voxel). Do voxels' tuning exhibit a single peak or multiple peaks? Does this differ from region to region? Could be useful

to plot the resulting tuning curves for a few sample voxels.

3. I found the term "directional tuning" confusing and potentially misleading. A suggestion would be to use "tuning" to indicate how focused the model weights are, and use another term (perhaps related to prediction, or to robustness) to indicate the correlation between predicted and actual time courses.

MINOR ISSUES:

4. Page 3, figure legend: "The inset depicts the median memory error": it is slightly confusing whether inset here refers to the little map on the top-right or the dots/box on the bottom-left.

5. Page 4, figure 2 caption: "Spacing and width were always matched to avoid over-representing certain directions". Not clear what is meant here. For clarity, I would suggest instead (or in addition) to mention the relevant passage from the Methods (i.e. "the broader each individual kernel, the fewer kernels were used").

6. Page 6, paragraph 2: the paper references Fig 1D, but this panel doesn't exist.

Reviewer #3:

Remarks to the Author:

Nau and colleagues analyzed a 7T fMRI dataset acquired while participants performed a navigation task in virtual environment. Participants memorized the locations of several everyday objects in a circular environment with visual cues on the walls, then were prompted to navigate towards indicated locations to return objects to their known position in the virtual world. The authors developed and applied an encoding model for head direction in the virtual environment and fit the model to voxel activation timecourses across the brain. The model allowed them to estimate the strength of direction-selective signals in each voxel, and to estimate the width of modeled tuning functions that best fit the observed activation. They observed reliable model fits in several areas previously identified as important for navigation and scene perception tasks (e.g., RSC, visual cortex, areas near hippocampus), and identified a gradient of tuning selectivity from narrow to broad tuning width along the posterior to anterior parahippocampus axis. Finally, they evaluated the behavioral relevance of these signals by comparing participants who performed well on the task (interpreted as those who learned quickly, and did not need to continue encoding locations throughout the study) to those who performed poorly (those who needed to continue learning locations). They found better model performance (tuning strength) and narrower tuning width for low-performance subjects than high-performance subjects, consistent with the notion that the low-performance subjects were still working to learn the environmental structure and improve their performance. Interestingly, disjoint sets of voxels were well-fit between the high- and low-performance groups, similar to previous demonstrations of a distinction between voxels responsible for scene perception vs scene construction. The authors conclude that his modeling effort supports a distributed code for heading direction that quantitatively differs across brain regions, and that this code reflects aspects of behavioral performance.

I think this is overall a strong report, with interesting analyses applied to an ecologically-valid task. I have a few concerns about how precise the analyses are/can be, and some ideas about how the authors could further reinforce their interpretation that these signals are most important when participants need to learn spatial features of an environment. I list several specific comments below.

Major:

1. Tuning width: a central goal of the article is to describe the tuning properties of different regions in

the human scene processing and navigation network. Part of this description involves identifying the 'tuning width' of the best-fit model to each voxel/across each ROI. The kernel-based modeling strategy they employ involves describing the activation in each voxel as a weighted combination of discrete direction-selective kernels (von Mises, tiled across the vHD space, with # of kernels scaling inversely with kernel width). They use a rigorous cross-validation procedure that involves separate steps for hyperparameter (λ) estimation and model selection, all of which I think is quite elegant. However, I'm wondering a bit about the ability of this modeling architecture to actually discriminate between different underlying tuning properties of each voxel (note that, of course, these approaches cannot speak to neural tuning properties, due to the intractable inverse problem relating neural and voxel-level signals). Let's consider an extreme example: what if the authors had used 360 kernels, each with very narrow tuning width? Such a model should be able to fully approximate the wider-kernel models (each wide kernel can be accurately described as a weighted combination of the tiny-kernels). While there are of course concerns with overfitting, etc, in this toy example, I think it illustrates the trickiness associated with model selection in this way. As such, I think the authors need to demonstrate (via simulations) that this modeling approach can accurately and unambiguously uncover underlying 'tuning' properties of each voxel across a range of SNR and tuning scenarios (unimodal voxel-level tuning; bimodal; random).

2. Model performance: related to the above, the authors are primarily comparing different versions of the same model architecture. But how does this type of model fare against, say, visual encoding models? Does this account for variance above and beyond other useful models, such as a motion energy model (Nishimoto et al, 2011)? And, related, can the authors demonstrate (again, via simulations), that there is no systematic bias associated with better model fits and wider/thinner tuning kernels? That is – under different 'true' voxel-level encoding models (unimodal thin vs thick tuning, bimodal tuning, random tuning), are there different expectations in terms of strength of model fit?

3. Effect of learning: can the authors make an effort to compare model fit properties for early and late halves of the experiment? I understand this makes the independent validation procedure tricky to implement, but even cross-validated model fitting here would be useful to bolster their argument that learning is more necessary for low-performance subjects, especially early in the experiment.

Minor:

1. Can the authors make some effort to describe the types of tuning they observe at the voxel level? Do voxels prefer individual directions? Multiple directions? Are some directions over-represented in the neural code compared to others?

2. Is it possible to include an example figure showing predicted and actual time-courses for an example voxel? (similar to the visual population receptive field papers; e.g., Dumoulin & Wandell, 2008)

3. The authors should describe the task in more detail – did the initial learning component occur inside the scanner? How is the fixation period modeled in the fMRI analyses? Where were the objects placed in the environment? What were the objects?

4. The variance explained seems to be extremely low (Supp Fig. 5 – Pearson's $r \sim 0.1$) – given these low values, how much could we expect to differentiate different versions of these models?

5. The authors mention a field map was acquired during imaging, but it's not clear how/if this scan was used during data (pre)processing

6. On pg 22 the authors describe a procedure where they exclude voxels that are not well-fit by the regularized encoding model (where a good λ could not be identified) – how many voxels were excluded?

Rebuttal letter for manuscript: “Behavior-dependent directional tuning in the human visual-navigation network”

We thank all reviewers for evaluating our manuscript (NCOMMS-19-32467-T) and for their very helpful comments, which we have addressed in detail below. In the attached manuscript and the Supplementary material file all changes are marked in red color. We believe that with the detailed and careful revisions outlined below the questions of the referees have been answered and that the revisions strengthened the conclusions of the manuscript.

Please note that in the course of the revisions we corrected two minor mistakes in the manuscript, which did not affect any conclusions.

- 1) The originally reported number of voxels in the early visual cortex regions of interest (ROI) mask was incorrect. Instead of the full ROI, we limited the analysis to calcarine grey matter voxels only.
- 2) The tuning width estimates of the posteromedial entorhinal cortex (pmEC) were switched with the ones of the anterolateral entorhinal cortex (alEC) in Figure 5D and Supplementary Figure 7D. This affected the visualization only, not the statistics. None of the two ROIs showed an effect of memory error on tuning width and all conclusions remain unchanged.

Reviewers' comments:

Reviewer #1 (Remarks to the Author):

Summary

In this manuscript the authors lay out a very interesting and novel approach to the problem of quantifying heading sensitivity in human fMRI data. In short, they develop an encoding model for heading direction and demonstrate that it explains variance in the responses of a wide array of visual areas during a navigation task. They then present the parameters of this model to help unpack the nature of heading representations within and the potential particular contributions of various cortical regions. The manuscript is well written, the question timely and interesting, the computational approach novel to this domain, and the presentation of the data laudably both clear and transparent. There are however a number of concerns that must be addressed.

We thank the reviewer for this encouraging summary of our manuscript and for evaluating it. We are particularly happy that it was timely, clear and transparent. The reviewer raises multiple important points in the comments regarding landmark processing, model selection and the effects of learning, which we address below.

Relationship between Brain and Behavior

By far the most problematic observation in this paper is that better behavioral performance is associated with more prevalent directional tuning in the EVC and worse performance with tuning in the medial parietal and temporal cortices. This result just doesn't make sense, and highlights potential problems with the behavioral task. The task is framed as requiring heading information, but heading here is completely confounded with the presence of particular landmarks. These landmarks are consistent, visible, and change size with distance from them. Could one explanation be, regardless of the laudable attempt to make their features similar, that participants are coding particular landmarks or their cooccurrence in the visual field? This confound would explain the apparent sensitivity to EVC to heading, and the differences between the results during stationary/turning and locomotion periods via differences in the changes in the landmarks appearance (see Supp 1). While it is possible that head direction encoding and low-level sensory processing are not mutually exclusive, the inability to separate these two processes is clearly a weakness of the study. How are the difference between the high and low memory error groups to be interpreted, even within the medial temporal lobe. An interesting experiment might be to remove the landmarks or even the entire arena during the participants response and see what of the apparent direction selectivity survives. Unfortunately, given the close spacing of the landmarks here, it seems unlikely that there is sufficient data when the landmarks were not visible or indicative of heading to analyze.

We thank the reviewer for raising these important points and apologize for the lack of clarity regarding how we think of directional tuning, what the sensory and neural origins might be and how it relates to behavior. We appreciate the opportunity to clarify these points here.

First, we would like to emphasize that the *“By far [...] most problematic observation [...] that better behavioral performance is associated with more prevalent directional tuning in the EVC”* apparently rests on a misunderstanding for which we apologize. Figures 5A-B, Supplementary Figure 6D-F, Supplementary Figures 8A-C as well as Supplementary Figures 9B-C show that behavioral performance is not related to directional tuning in the EVC on group level. Instead, we observed this relationship for more lateral and ventral occipital lobe regions (Figure 4), but most strongly for higher order areas such as the parahippocampal cortex as well as regions in the medial temporal lobe (e.g. Figure 4 & 5). To clarify this point, we added a new Supplementary Figure 6H showing that there is no correlation between model performance and memory error in the EVC and emphasized it in the text.

New Supplementary Fig. 6G-H

Caption: G-H) Scatter plot of posteromedial entorhinal cortex (pmEC) and early visual cortex (EVC) model performance at group-optimal tuning width of 45° over across-trial median memory error (Fig. 1). Least-square line as well as permutation-based rank & linear-correlation coefficients were added. There is a correlation between model performance and memory error in pmEC, but not in the EVC.

Second, and on a more general level, we believe a potential source of confusion here might have been the lack of a clear definition of the term “directional tuning” and what spatial information might be used by the brain to infer it. The only information participants have to infer direction from in this task are the landmarks. Removing those would make orienting and hence the task impossible. We therefore would not expect to see directional fMRI-tuning anywhere in the brain in the total absence of these landmarks, and that the tuning fades quickly after landmark removal due to path integration error accumulation. Indeed, if participants remained stationary after landmark removal, the tuning should remain stable for a limited amount of time. This possibility is exciting, and we plan to address it in future work, but it does neither validate nor invalidate any claims we make in the present manuscript.

The reviewer points out a “confound” between landmark occurrence and direction, eluding to the facts that some landmarks might be over-/undersampled while walking into certain directions, that not all landmarks can be seen while walking in all directions and that there are always landmarks in the scene. Naturally, while walking North the landmarks to the South are out of sight. The number of landmarks, their size and combination however depend on their distance to the observer, which is not direction-specific due to our careful matching of low-level visual features of our landmarks. Hence, even if participants saw some landmarks more often on average while walking in some directions versus others, the only difference between these directions is the identity of the landmark. As the reviewer pointed out correctly, such “coding [of] particular landmarks” might contribute to the directional tuning we observe, it is however also explicitly what has been suggested necessary to anchor directional representations to the visual scene (see for example Skaggs et al. 1995, Vann et al. 2009, Evans et al. 2016, Bicanski et al. 2018 or Page & Jeffery 2018). Somewhere along the cortical hierarchy, this landmark-based code is converted into a non-sensory head direction code. The point of our manuscript is to investigate brain networks that engage in this inference of directional information from the environment, test how the underlying tuning differs between different parts of the cortical hierarchy, and to examine how all of this depends on behavior. Critically, we do not claim that we are measuring activity related to head direction cells, or that the presented results are in any way independent of visual input. We clarified this point in the manuscript as described later.

Please also note that the manuscript now includes several new analyses that might help to address this point further. One of the analyses presented in our new Supplementary figure 5 depicts the model weights across directions for individual voxels and ROIs. Here, we show that different voxels have a different tuning profile, a finding which also supports the view that low-level visual or sampling confounds do not explain our results. If they did, the same low-level biases should have enforced the same tuning profile onto all voxels at least in the early visual cortex, which is not what we see. As pointed out by the reviewer, individual voxels might still encode the presence of particular landmarks, but we do not claim that they do not, and this might exactly be the information the brain uses to infer direction in the first place.

In another newly added analysis (Supplementary Figure 5C), we performed leave-one-run-out cross-validation to examine the directional tuning in each individual scanning run. Again, we saw no effect of memory error on the model performance in the EVC, not even when cross-validated across runs. Interestingly, these analyses revealed further differences in tuning between the EVC, the parahippocampal cortex and the entorhinal-hippocampal-circuit. While the model performance in EVC did not depend on memory error, it did so in the parahippocampal cortex with a remarkable stability across runs. Likewise, the tuning in the entorhinal-hippocampal circuit also reflected memory error, but unlike in the parahippocampal cortex the relationship between these variables seemed to develop over time. This further emphasizes that the tuning in the EVC is overall drastically different from

the one in higher order areas and that the relationship to the memory error arises at a higher level of the cortical hierarchy, not at the visual input stage.

Finally, and regarding the reviewer's question about the interpretation of the presented (model x memory performance) effects, we thoroughly discuss the results in the Discussion and provide multiple possible explanations. In short, we believe the difference in model performance in higher order areas between the participant groups indicates that these participants follow a different cognitive strategy, and/or that the tuning reflects the state of the formation of a cognitive map (i.e. how much spatial knowledge participants encode). If so, these two possibilities likely interact (i.e. if a participant does not form a cognitive map of the environment, or has already done so successfully, we do not expect to see tuning or effects of learning on this process).

Any mechanistic explanation of this phenomenon remains speculation due to the population-level resolution of fMRI. One possible explanation could be the following. Because fMRI likely measures synaptic processing (see e.g. work by the Logothetis lab), the difference in tuning in these higher order areas might reflect how much input these areas "recruit". The MTL of high-error participants might receive stronger inputs from perceptual regions, because these participants are still in the process of forming a cognitive map, which heavily relies on visual information. Accordingly, low-error participants might have formed a cognitive map already and require less inputs to the MTL from perceptual regions. Another related explanation could be that the tuning reflects the state of memory consolidation more directly, meaning that it could reflect how much information is being encoded at every given time. This is consistent with the observed continuous disengagement of the hippocampus in the course of spatial learning (e.g. Brodt et al. 2016). As pointed out above, this likely interacts with the cognitive strategies the participants follow, which should be assessed in more detail in future experiments.

To summarize, we fully acknowledge (and discuss in the manuscript) the fact that we do not disentangle visual from non-visual virtual head direction representations, but it is also not the intention of this work. Here, we show that we can study directional tuning during spatial VR-navigation in human fMRI, that we can map its tuning width for the first time, that the tuning differs across regions and that it relates to spatial memory performance and locomotion. The directional information used for the underlying computations necessarily comes from the visual landmarks. We discuss the possible explanations of our results and the limitations of this work thoroughly and transparently in the Discussion section and make several predictions to guide future work. To clarify the above-mentioned points, we adapted multiple sections of the manuscript, added new analyses and Supplementary Figures and now provide a better definition of the terms used as described below.

Following changes were made to the manuscript (underlined):

Page 2

“[...] we then developed an iterative kernel-based encoding model (Fig. 2) of the participants’ navigation behavior (Supplementary Fig. 1) to map directional tuning across the human cortex. In this framework, voxels are considered to be directionally tuned if their activity can be predicted based on world-centered virtual head direction (vHD), i.e. the direction a participant is facing within the virtual arena at each moment in time.”

Page 4

“We define directional tuning strength as the model performance, i.e. how well the model predicted the time course of a voxel or region. Critically, a positive tuning strength suggests that a voxel selectively represents some vHD’s over others in a temporally stable manner.”

Page 6

Visualizing the model weights showed that different voxels preferred different directions and that there were no distinct tuning profiles (uni-, bi-, trimodal weight distributions) observable on ROI level.

Page 9

We performed leave-one-run-out cross-validation of our full modeling pipeline to obtain the directional tuning strength for each ROI not only for our original test run but for all runs. We found three main patterns of results (Supplementary Fig. 8C): EVC and RSC tuning strength again did not reflect memory performance, also not when cross-validated over runs. In contrast, PHG tuning strength did reflect memory performance robustly over runs. Strikingly, HPC and EC tuning also reflected memory performance, but the relationship between the two variables developed over time.

Page 13

We also observed directional tuning in the early visual cortex for which no HD-cells have been reported to date. This raises the question whether the effects reported here are due to HD, lower-level sensory or landmark processing? We believe these options are not exclusive and the underlying processes might strongly interact in our naturalistic task. The only source of information to infer direction here were the landmarks, whose visibility at every moment depended on the position and the direction of the participant. Importantly, even if some landmarks were seen more often than others while walking in a certain direction, because they were all matched in low-level visual features the only difference between directions was landmark identity. Such landmark processing and the integration of landmarks across viewpoints has been suggested to be fundamental to anchor our sense or direction in space (Yoder et al., 2011). In addition, locomotion (Ayaz et al., 2013; Saleem et al., 2013) and world-centered location (Fournier et al., 2019; Saleem et al., 2018) have been shown to modulate EVC activity in rodents.

New reference

Yoder, R. M., Clark, B. J. & Taube, J. S. Origins of landmark encoding in the brain. Trends Neurosci. 34, 561–571 (2011).

Supplementary Figure 5

A) Model weights across directions for exemplary voxels & ROIs

Caption: Tuning profiles for randomly selected sample voxels and regions of interest (ROIs). We plot the model weights across directions for one exemplary voxel of each ROI (left panel) and the model weights averaged across voxels of these ROIs (middle panel). To test whether there are tuning profiles that were consistent over voxels (e.g. uni-, bi-, trimodality) but averaged out across voxels, we additionally aligned the peak model weight across voxels (right panel). We plot the mean (solid line) and one standard deviation (shaded area) across voxels. ROIs: early visual cortex (EVC), retrosplenial cortex (RSC), parahippocampal gyrus (PHG), hippocampus (HPC) and posteromedial entorhinal cortex (pmEC). This figure shows that different voxels have distinct tuning profiles and prefer different directions.

Page 15

Taken together, low-level sensory information alone can explain neither the tuning we observed, nor its relationship to behavioral performance. Instead, our results are well in line with previous work on higher-level spatial representations and with landmark processing in the visual system. We did not disentangle visual from non-visual factors here, but the tuning in the EVC and its progression over time clearly differed from the one in MTL regions. The relationship between directional tuning and spatial memory hence arose on a higher level of the cortical hierarchy, not the visual input stage. However, our results support the notion that these higher-level cognitive processes are closely intertwined with visual processing up to its earliest stages in the brain.

Page 17

Looking forward, we see a wide range of future applications and exciting avenues to be explored. [...] In addition, the influence of different sensory and behavioral variables on the present results could be further examined by removing the landmarks and testing which areas maintain their tuning, or by comparing the results to the ones of other encoding models such as one of motion energy (Nishimoto et al., 2011).

Supplementary Fig. 8C

Model performance across scanning runs

Caption: Model performance (see Fig. 5C) across scanning runs. We performed leave-one-run-out cross-validation for voxels within the ROIs to examine how directional tuning develops over time. In each cross-validation loop a different scanning run was taken as test run, while all others served as training runs. We plot the model

performance (Z-score) for each run for high and low memory error participants (median-split by memory error) for each ROI in two formats: 1) mean (solid line) and standard error of the mean (shaded area) for each run, as well as 2) the average model performance across runs as group-level whisker-boxplots.

Tuning Width Analysis

The analysis of tuning width seems to suffer from a small circularity. While the basic fitting of the model is well validated in independent data, the tuning width is simply taken as the model that produced the best fits. Even though each of the models that is input into this selection is fair, selecting the best of the eight still introduces a significant bias. Here is some matlab code that illustrates the problem in random data that has no difference between the conditions:

```
randSet = rand(100,8,10000) >= .25;
allResult = squeeze(mean(randSet));
bestResult = squeeze(max(allResult));
avgResult = squeeze(mean(allResult));
[mean(avgResult) mean(bestResult)]
figure; histogram2(bestResult, avgResult); axis([.65 1 .65 1 0 Inf]); xlabel('Best'); ylabel('Avg');
```

To be clear, this problem likely does not detract significantly from many of the author's arguments but should be removed wherever possible, though this is sometimes tricky. For example, in Figure 3A, the depicted T-values are from the best model, but should be from the average, the currently presented values are inflated. In Figure 3B, the tuning width presented is the average from independent circular selections of tuning widths, making the average likely okay to present, but not test is run to show that the T-values for the presented average widths are actually better than the other widths. In Figure 4C, significance markings are plotted against zero for the model performance, but these z-scores are artificially inflated and no such test should be performed. In Figure 4D, the variability amongst the participants in the tuning width metric is apparent, which makes some of the claims a bit difficult to rely on heavily. In general, I recommend that the authors carefully evaluate how best to present and analyze this metric and attenuate their claims about it.

The reviewer here points out a "small circularity" in our tuning width analysis, suggests specific changes to our model selection and recommends attenuating our claims regarding tuning width. We appreciate this constructive and creative feedback and thank the reviewer for it. The model selection is indeed performed on the testing run and relies on a "winner-takes-all" logic. We chose this model selection purposefully and believe it is not easy to be replaced for multiple reasons.

First and most importantly, the present work suggests that the directional tuning width of a region (such as the retrosplenial cortex, see Fig. 5) might depend on behavioral performance. If we performed the model selection in any other run but the one tested, we

would likely select the “wrong” model for this run, simply because the behavioral performance might be different. Looking forward, we believe this will be a general challenge for studying the dynamics of a certain behavioral/sensory tuning going beyond the present work.

Second, to implement our width mapping procedure we took direct inspiration from classical population receptive field (pRF) mapping, one of the most commonly used encoding models in the neuroimaging field of vision. All pRF-models are typically tested on the same data of a voxel until the best fitting model is determined. To map pRF-parameters such as ‘size’ across the brain, these ‘size maps’ are often thresholded based on how much variance was explained by the winning model. This is not in principle different than the model selection presented here and is a common practice in the field. As the reviewer pointed out, our approach is even more stringent than traditional pRF-mapping procedures, because we estimate the weights used for the final prediction on independent data.

Third, we considered other selection criteria such as taking the mean model instead of the best model as suggested by the reviewer, this however has other (and from our perspective outweighing) disadvantages. The goal of the iterative model fitting is to find the parameters that best explain the data. Necessarily, these parameters must be independent of how well all other parameters performed. In an extreme case, if we tested e.g. 1000 models of which the majority was necessarily wrong, taking the mean across all of those would completely overshadow even that one model that potentially perfectly fit the data. Taking the average would therefore introduce other more problematic issues, because it depends on the types and the number of models tested.

The following example illustrates this problem.

```
% Hypothetical model performance scores / correlations  
r = [1, 20, 60]/100;
```

```
% Model selection  
[~, win_mean] = min(abs(r-mean(r))); [~, win_max] = max(r);
```

The best model would be model 3, the mean model would be model 2. However, if we instead tested 100 models, including the three from above, the best model would still be number 3, but now the mean model would be model 1.

```
% get model performance & perform model selection  
r = [1, 20, 60, ones(1,97)]/100; [~, win_mean] = min(abs(r-mean(r))); [~, win_max] = max(r);
```

Taking the best model has at least one strong advantage: it is invariant to how well the other models perform.

Forth, we think the comparison of tuning width between participant groups the reviewer refers to is fair, because the same model selection was performed on both groups. Even if there were any biases, the same biases affected the two groups equally and should not interfere with the comparison. The group variability in Figure 4D is high indeed, but the used statistical (permutation-based) test is taking the data distribution into account.

Finally, we acknowledge that our model selection might not be perfect in all regards (like any other model selection), but believe the combination with our strict cross-validation procedure represents the best compromise between robustness and sensitivity to detect the behavior-dependent effects on the directional tuning we observed. The alternatives would have limitations of their own, which in our view outweigh the present ones. In addition, similar encoding model work such as pRF-mapping builds on a similar model selection procedure, which is routinely used by many labs and represents a standard in the field. However, we comply with the reviewers suggestion to tone down the claims made based on the tuning width metric. In addition, we point the reader to the question of model selection transparently as described below.

Following changes were made to the manuscript (underlined):

Page 8

[...] the tuning width differed between groups ($t(18) = 2.04$, $p = 0.044$). This suggests that participants with high memory error might show a sharper tuning in RSC than participants with low memory error (Fig. 5D).

Page 11

[...] it was the tuning width and topology in RSC that depended on memory, notably however with a large variability across participants.

Page 25

Again, we iterated this analysis for all basis sets (Fig. 5B), yielding the tuning strength (model performance) and the tuning width (the width of the vHD-kernels in the basis set) of each ROI in the test set. While other model selection procedures are possible, ours enabled to estimate the tuning properties of a given region in a given run even if the tuning underwent changes across runs.

Learning Slope Analysis

I am somewhat confused by the purpose of this analysis. The high-memory error group has room to improve and the low-memory error group does not. Why is it surprising that the slopes differ? Is there some specific point to this quantification?

We apologize for the lack of clarity and thank the reviewer for the opportunity to elaborate more on this point here. One of our key messages of the manuscript is that the directional tuning we observe depends on the behavior participants express. Memory error is the most important behavioral read-out in this context. As the reviewer pointed out, the purpose of

this and the related memory error analyses was indeed to test whether high and low memory error groups approach the same level of performance.

This might seem trivial but did not necessarily have to be the case. Our neuroimaging analyses (Fig. 4, 5) showed stronger tuning in medial temporal lobe regions in the high- compared to the low-error participants, an effect that was surprising to us at first. As discussed above, one explanation could have been that the directional tuning in these regions reflects memory encoding/learning. If true, this suggested that participants with high error (strong tuning) are still in the process of learning, while participants with low error already reached memory ceiling (and hence stopped learning). To test this prediction, we analyzed the learning slopes of our participants, which confirmed that a) both participant groups approached a similar level of performance and b) that the high memory error group therefore had steeper learning slopes.

Please note that the newly added leave-one-run-out cross-validation procedure provides new insights into these learning effects. In short, we show that the directional tuning is a) robust even when cross-validated across scanning runs, b) does not differ between groups in the early visual- and the retrosplenial cortex, but in the parahippocampus and the medial temporal lobe and c) that the latter also shows changes in this relationship between behavior x model performance across runs. In response to this comment (overlapping with comment 1 by reviewer 2 and comment 3 by reviewer 3), we adapted the paragraph that explained the logic behind the learning slope analysis significantly and discuss the learning effects in more detail as described below.

Following changes were made to the manuscript (underlined):

Page 9

Contrary to our initial hypothesis, the differences between groups could reflect general differences in the cognitive strategy used or the participants' ongoing effort in encoding rather than retrieving a map of the environment. To investigate these effects post-hoc in more detail we examined how directional tuning strength developed in the course of the experiment. We performed leave-one-run-out cross-validation of our full modeling pipeline to obtain the directional tuning strength for each ROI not only for our original test run but for all runs. We found three main patterns of results (Supplementary Fig. 8C): EVC and RSC tuning strength again did not reflect memory performance, also not when cross-validated over runs. In contrast, PHG tuning strength did reflect memory performance robustly over runs. Strikingly, HPC and EC tuning also reflected memory performance, but the relationship between the two variables developed over time. Over scanning runs, tuning strength tended to increase in high-error participants and tended to decrease in low-error participants. Interestingly, on a behavioral level, we found that both groups approached the same level of memory performance in the course the experiment (Supplementary Fig. 2C-D), but that the low-memory-error group had approached this performance level earlier in the experiment than the high-memory-error group (Supplementary Fig. 2D-E).

Page 12

“The direction of this effect and how it developed over time in MTL regions speaks to the idea that the two participant groups might follow different cognitive strategies and that the tuning reflects how well the environment has been encoded.”

Page 16

Hypothetically, because fMRI likely measures synaptic processing rather than output spiking (Logothetis et al., 2001), the tuning could reflect how much inputs a region receives. For example, the MTL of high-error participants might receive stronger inputs from perceptual regions, because these participants were still in the process of forming a cognitive map, which heavily relies on visual information. Accordingly, low-error participants might have formed a cognitive map already. Consistent with this idea, we found that both participant groups approached the same level of memory performance, but that participants with stronger tuning in higher-level regions showed steeper learning curves than those with weaker tuning (Supplementary Fig. 2C). This is also consistent with earlier reports showing that hippocampal activity tracks the amount of knowledge obtained at a given time rather than the accumulated absolute knowledge (Wolbers and Büchel, 2005) and with decreases in hippocampal activity in the course of spatial learning (Brodt et al., 2016).

New reference:

Brodt, S., Pöhlchen, D., Flanagin, V.L., Glasauer, S., Gais, S., and Schönauer, M. (2016). Rapid and independent memory formation in the parietal cortex. Proc. Natl. Acad. Sci. 113, 13251–13256.

Page 16

Analyzing how the tuning developed over time revealed that such continuous learning-related disengagement could however explain only the results observed for the low-, not of the high-memory-error group. The latter even showed slight increases in directional tuning over time in the entorhinal cortex (Supplementary Fig. 8C). One account that would reconcile these findings is that the memory-dependent effects could reflect a difference in cognitive strategy used, which might develop and get refined in the course of spatial learning.

Page 26

“To examine how directional tuning developed over time, we further performed leave-one-run-out cross-validation across all scanning runs (Supplementary Fig. 8C). We performed the ROI-level encoding modeling analysis (Fig. 5) for all runs, each time estimating the model weights on all runs but the one used for testing.”

Supplementary Fig. 8C

Model performance across scanning runs

Caption: Model performance (see Fig. 5C) across scanning runs. We performed leave-one-run-out cross-validation for voxels within the ROIs to examine how directional tuning develops over time. In each cross-validation loop a different scanning run was taken as test run, while all others served as training runs. We plot the model performance (Z-score) for each run for high and low memory error participants (median-split by memory error) for each ROI in two formats: 1) mean (solid line) and standard error of the mean (shaded area) for each run, as well as 2) the average model performance across runs as group-level whisker-boxplots.

Following statement on page 16 was removed:

“In the present study, we did not investigate how directional representations develop over time.”

Small Comments

1. The presentation of the memory error as essentially arbitrary units is unnecessarily opaque. These can be converted to proportions of the total arena diameter to give a better sense of the extent of the mislocalizations and the improvements over trials.

We thank the reviewer for suggesting a more intuitive unit to present memory performance. We agree that this is helpful here and now report the memory error in “percent of arena size” in Fig. 3 and Supplementary Fig. 2.

2. I couldn’t find an explicit statement of what the various significance markings (*, **) corresponded to in the figure legends.

We thank the reviewer for pointing us to missing figure legends for which we apologize. The description of the significance markings has been added to the captions of Figure 5 and 6 as follows:

Figure 5 & Figure 6

“Permutation-based t-test results were added: **p < 0.05, FDR-corrected, *p < 0.05, uncorrected”.

Reviewer #2 (Remarks to the Author):

In this manuscript, Nau and colleagues investigate directional tuning in the human brain using 7T-fMRI and a virtual navigation task. Building on previous successes of encoding models in the orientation selectivity domain, the authors find directional tuning across several regions of the ventral occipital, medial temporal, and medial parietal lobes. They also report that voxel time courses in mnemonic regions could be better predicted when subjects performed poorly (!) on a spatial memory task, and that tuning in several regions depended on whether subjects were moving.

The manuscript is technically strong, extremely clear, and very well written. My main critique is that, having established the effectiveness of this encoding model approach, the study misses on the opportunity to deliver more novelty such as delving deeper on the links to behavior. To be clear, I believe the manuscript is very strong and there are no major flaws, so the comments below are only a few suggestions of how I believe it could be made stronger.

We thank the reviewer for evaluating our manuscript, the positive and encouraging comments and for raising important questions about what “directional tuning” is and how it changes in the course of learning. Please find our detailed responses to the individual comments below.

MAJOR COMMENTS:

1. The authors hypothesize that "tuning should be stronger in ventral visual stream and medial temporal regions in participants that performed well in the spatial memory task". Instead, they found that tuning was stronger in those regions in participants with high memory error. They speculate that the observed group differences could be explained by the fact that learning takes longer in participants with high memory error. I think that this is an interesting hypothesis and could be explored further. For instance, could be authors test whether directional tuning reduces over time (e.g. looking at the correlation between predicted and observed time courses either on a voxel-by-voxel or ROI analysis)? If it does, does it reduce more quickly in participants with low memory error?

The reviewer here asks the interesting question about how the tuning changes with learning and suggests assessing the model performance over time. To address it (as well as the related comment 3 by reviewer 3), we implemented an additional leave-one-run-out cross-validation procedure on ROI-level that enables visualizing how the tuning develops in the course of the experiment.

The analysis is analogous to the one presented in the original manuscript, but now includes

an “outer cross-validation loop” that iterates across runs for model testing. For each iteration, we train the model on all runs but one, and test it on the one left out. We visualized the results in the newly added Supplementary Fig. 8C.

First, this analysis showed that our key effects of interest survived the cross-validation procedure, showing that the originally reported relationship between memory error and directional tuning is robust. Early visual and retrosplenial cortex had a positive directional tuning strength in all runs, and the tuning did not differ between high- and low-error participants. In contrast, parahippocampal and entorhinal cortex tuning again reflected the relationship to memory error. Interestingly, when averaged across the experiment, also the hippocampus showed tuning only in the high-memory group, like entorhinal and parahippocampal regions. These results are important, because they show that one of the key messages of the manuscript (i.e. the directional tuning relates to behavior) is robust even when cross-validated across all data partitions.

More interesting however could be the differences in tuning between the individual scanning runs as eluded to by the reviewer. Our new analysis speaks directly to this question and shows that the tuning strength in early visual cortex tended to increase in the course of the experiment. This could potentially reflect that the viewing behavior becomes more stereotypical as the environment and task become more familiar. As outlined in the discussion of the original manuscript, future experiments using eye tracking could test this prediction specifically.

In the retrosplenial cortex, tuning strength varied slightly over time, but did not show a clear up- or downward trend across the experiment. In contrast, parahippocampal tuning strength and its difference between high- and low-error groups remained extraordinarily stable. The tuning strength in the high-error group remained elevated, the one in the low-error group remained basically absent. Please note that because both high and low memory error groups roughly approach the same level of memory performance in the course of the experiment, these new results suggest (to us) that the underlying memory-related parahippocampal computation does not change during learning. Possibly, it reflects a difference in cognitive strategy between the groups.

Strikingly, the entorhinal cortex as well as the hippocampus showed yet a different pattern of results. While the low-error group showed a decrease in tuning strength, the high-error group even showed a slight increase over time. This further points to the idea that these two participant groups might follow distinct cognitive strategies, which even get refined in the course of learning.

Importantly, the fact that there is no difference in tuning strength between groups in the early visual cortex and the retrosplenial cortex, and that the difference in the parahippocampus remains stable over time suggests that the changes in tuning we observe in the entorhinal cortex and the hippocampus do not reflect biases in behavior or low-level

visual inputs per se. The difference in directional tuning observed here and hence the relationship to memory performance clearly arises at a higher level of the cortical hierarchy. We added these new results to Supplementary Figure 8 and discuss it in the main text as described below.

Following changes were made to the manuscript (underlined):

Page 9

Contrary to our initial hypothesis, the differences between groups could reflect general differences in the cognitive strategy used or the participants' ongoing effort in encoding rather than retrieving a map of the environment. To investigate these effects post-hoc in more detail we examined how directional tuning strength developed in the course of the experiment. We performed leave-one-run-out cross-validation of our full modeling pipeline to obtain the directional tuning strength for each ROI not only for our original test run but for all runs. We found three main patterns of results (Supplementary Fig. 8C): EVC and RSC tuning strength again did not reflect memory performance, also not when cross-validated over runs. In contrast, PHG tuning strength did reflect memory performance robustly over runs. Strikingly, HPC and EC tuning also reflected memory performance, but the relationship between the two variables developed over time. Over scanning runs, tuning strength tended to increase in high-error participants and tended to decrease in low-error participants. Interestingly, on a behavioral level, we found that both groups approached the same level of memory performance in the course the experiment (Supplementary Fig. 2C-D), but that the low-memory-error group had approached this performance level earlier in the experiment than the high-memory-error group (Supplementary Fig. 2D-E).

Page 12

"The direction of this effect and how it developed over time in MTL regions speaks to the idea that the two participant groups might follow different cognitive strategies and that the tuning reflects how well the environment has been encoded."

Page 16

Hypothetically, because fMRI likely measures synaptic processing rather than output spiking (Logothetis et al., 2001), the tuning could reflect how much inputs a region receives. For example, the MTL of high-error participants might receive stronger inputs from perceptual regions, because these participants were still in the process of forming a cognitive map, which heavily relies on visual information. Accordingly, low-error participants might have formed a cognitive map already. Consistent with this idea, we found that both participant groups approached the same level of memory performance, but that participants with stronger tuning in higher-level regions showed steeper learning curves than those with weaker tuning (Supplementary Fig. 2C). This is also consistent with earlier reports showing that hippocampal activity tracks the amount of knowledge obtained at a given time rather than the accumulated absolute knowledge (Wolbers and Büchel, 2005) and with decreases in hippocampal activity in the course of spatial learning (Brodt et al., 2016).

New reference:

Brodt, S., Pöhlchen, D., Flanagin, V.L., Glasauer, S., Gais, S., and Schönauer, M. (2016). Rapid

and independent memory formation in the parietal cortex. *Proc. Natl. Acad. Sci.* *113*, 13251–13256.

Page 16

Analyzing how the tuning developed over time revealed that such continuous learning-related disengagement could however explain only the results observed for the low-, not of the high-memory-error group. The latter even showed slight increases in directional tuning over time in the entorhinal cortex (Supplementary Fig. 8C). One account that would reconcile these findings is that the memory-dependent effects could reflect a difference in cognitive strategy used, which might develop and get refined in the course of spatial learning.

Page 26

“To examine how directional tuning developed over time, we further performed leave-one-run-out cross-validation across all scanning runs (Supplementary Fig. 8C). We performed the ROI-level encoding modeling analysis (Fig. 5) for all runs, each time estimating the model weights on all runs but the one used for testing.”

Supplementary Fig. 8C

Model performance across scanning runs

Caption: Model performance (see Fig. 5C) across scanning runs. We performed leave-one-run-out cross-validation for voxels within the ROIs to examine how directional tuning develops over time. In each cross-validation loop a different scanning run was taken as test run, while all others served as training runs. We plot the model performance (Z-score) for each run for high and low memory error participants (median-split by memory error) for each ROI in two formats: 1) mean (solid line) and standard error of the mean (shaded area) for each run, as well as 2) the average model performance across runs as group-level whisker-boxplots.

Following statement on page 16 was removed:

“In the present study, we did not investigate how directional representations develop over time.”

2. What is the tuning profile of a "typical" voxel (if there's such a thing as a typical voxel). Do voxels' tuning exhibit a single peak or multiple peaks? Does this differ from region to region? Could be useful to plot the resulting tuning curves for a few sample voxels.

The reviewer here raises the intriguing question about the nature of the tuning profile of typical voxels and regions. This is important because voxels might represent a single direction or multiple ones, which not only affects how we think about the underlying neural process but could also lead to further discoveries (e.g. distinct tuning profiles for different regions).

To address this question (related to 'minor' comment 1 by reviewer 3), we visualized the model weights across directions for a representative voxel in each region of interest (ROI) of an exemplary participant as well as the across-voxel-average of these weights for each ROI. Because the represented directions might differ across voxels but still follow a similar tuning profile (e.g. bimodality), we additionally aligned the peak-direction (i.e. the one with the highest weight) across voxels before averaging. If voxels shared a similar tuning profile, this should be observable in this latter analysis. The results are depicted in our new Supplementary Fig. 5.

We observed that "typical" voxels tended to represent multiple directions and that there was no striking common tuning profile emerging across voxels or regions (i.e. no uni-, bi- or trimodality). Instead, the model weight distribution differed across voxels, averaging out on ROI-level. This was true even after aligning the peak directions across voxels, again demonstrating that not only the represented directions but also the whole tuning profile differed. Please note that we plot the weights corresponding to a tuning width of 10° here to visualize the tuning profile with our highest resolution.

Importantly, the fact that the model weights differ across voxels emphasizes that there were no major imbalances in sampling or visual similarity across directions either. If there were, this should have led to a systematic bias in model weights at the very least in the early visual cortex, which is not what we see. In a future experiment, it could be interesting to have more distinct landmarks in the distance and try to triangulate from the model weights which landmarks had the strongest effect (to study landmark anchoring).

Following changes were made to the manuscript (underlined):

Page 6

Visualizing the model weights showed that different voxels preferred different directions and that there were no distinct tuning profiles (uni-, bi-, trimodal weight distributions) observable on ROI level (Supplementary Fig. 5).

Supplementary fig. 5

A) Model weights across directions for exemplary voxels & ROIs

3. I found the term "directional tuning" confusing and potentially misleading. A suggestion would be to use "tuning" to indicate how focused the model weights are, and use another term (perhaps related to prediction, or to robustness) to indicate the correlation between predicted and actual time courses.

We are grateful to the reviewer for pointing out that the term “directional tuning” might be confusing for some readers, apologize for not defining this term more clearly and appreciate the opportunity to do so now. It is a term that is used often in the manuscript, and that even appears in the title, and a clear understanding of what it means is critical to appreciate our manuscript as a whole. As a first step in addressing this comment, we added following statements to the manuscript (underlined).

Following changes were made to the manuscript (underlined):

Page 2

“[...] we then developed an iterative kernel-based encoding model (Fig. 2) of the participants’ navigation behavior (Supplementary Fig. 1) to map directional tuning across the human cortex. In this framework, voxels are considered to be directionally tuned if their activity can be predicted based on world-centered virtual head direction (vHD), i.e. the direction a participant is facing within the virtual arena at each moment in time.”

Page 4

“We define directional tuning strength as the model performance, i.e. how well the model predicted the time course of a voxel or region. Critically, a positive tuning strength suggests that a voxel selectively represents some vHD’s over others in a temporally stable manner.”

Second, we would also like to use this opportunity to underline that we do not refer to “directional tuning” in isolation, but instead introduced two key terms that describe this tuning in more detail. On page 4, we defined the term “Tuning strength” as:

Page 4

“[...] the model performance, i.e. how well the model predicted the time course of a voxel or region”.

In addition, we defined the “Tuning width” as:

Page 4

“[...] the FWHM of the kernels leading to the best model performance”.

We believe especially the term “tuning strength” resembles the reviewer’s suggestion and offers a clearly defined term to describe the correlation between predicted and actual time courses. Importantly, we do not think of the predictability per se as “tuning”, but that the fact that we see a successful prediction in a voxel also suggests that this voxel is directionally tuned. We therefore argue that using the term “directional tuning” broadly is still valid here, even though assessing the predictive potential of the model weights might be an indirect way of measuring it. Related to the reviewer’s comment 2, we now also visualize the model weights in our new Supplementary Fig. 5 and hence provide a more thorough description of the tuning profiles. Together with the newly added definitions and clarifying statements, we hope to convince the reviewer that using the term directional tuning is valid in the context of our manuscript.

MINOR ISSUES:

4. Page 3, figure legend: "The inset depicts the median memory error": it is slightly confusing whether inset here refers to the little map on the top-right or the dots/box on the bottom-left.

We appreciate the reviewer's suggestion and clarified this ambiguity in the figure caption. It now reads as follows.

Figure 1

"The inset on the bottom left depicts the median memory error across trials for single participants and as whisker-boxplot"

5. Page 4, figure 2 caption: "Spacing and width were always matched to avoid over-representing certain directions". Not clear what is meant here. For clarity, I would suggest instead (or in addition) to mention the relevant passage from the Methods (i.e. "the broader each individual kernel, the fewer kernels were used").

We appreciate also this suggestion and added following statement to the figure caption.

Figure 2

"Spacing and width were always matched to avoid over-representing certain directions, i.e. the broader each individual kernel, the fewer kernels were used."

6. Page 6, paragraph 2: the paper references Fig 1D, but this panel doesn't exist.

We thank the reviewer for pointing us to this mistake, which we corrected.

Reviewer #3 (Remarks to the Author):

Nau and colleagues analyzed a 7T fMRI dataset acquired while participants performed a navigation task in virtual environment. Participants memorized the locations of several everyday objects in a circular environment with visual cues on the walls, then were prompted to navigate towards indicated locations to return objects to their known position in the virtual world. The authors developed and applied an encoding model for head direction in the virtual environment and fit the model to voxel activation timecourses across the brain. The model allowed them to estimate the strength of direction-selective signals in each voxel, and to estimate the width of modeled tuning functions that best fit the observed activation. They observed reliable model fits in several areas previously identified as important for navigation and scene perception tasks (e.g., RSC, visual cortex, areas near hippocampus), and identified a gradient of tuning selectivity from narrow to broad tuning width along the posterior to anterior parahippocampus axis. Finally, they evaluated the behavioral relevance of these signals by comparing participants who performed well on the task (interpreted as those who learned quickly, and did not need to continue encoding locations throughout the study) to those who performed poorly (those who needed to continue learning locations). They found better model performance (tuning strength) and narrower tuning width for low-performance subjects than high-performance subjects, consistent with the notion that the low-performance subjects were still working to learn the environmental structure and improve their performance. Interestingly, disjoint sets of voxels were well-fit between the high- and low-performance groups, similar to previous demonstrations of a distinction between voxels responsible for scene perception vs scene construction. The authors conclude that his modeling effort supports a distributed code for heading direction that quantitatively differs across brain regions, and that this code reflects aspects of behavioral performance. I think this is overall a strong report, with interesting analyses applied to an ecologically-valid task. I have a few concerns about how precise the analyses are/can be, and some ideas about how the authors could further reinforce their interpretation that these signals are most important when participants need to learn spatial features of an environment. I list several specific comments below.

We thank the reviewer for this positive evaluation of our manuscript, the constructive feedback regarding our modeling framework and for pointing us to parts of the manuscript in which relevant information was missing. Please find our responses to the specific comments below.

Major:

1) Tuning width: a central goal of the article is to describe the tuning properties of different regions in the human scene processing and navigation network. Part of this description involves identifying the ‘tuning width’ of the best-fit model to each voxel/across each ROI. The kernel-based modeling strategy they employ involves describing the activation in each voxel as a weighted combination of discrete direction-selective kernels (von Mises, tiled across the vHD space, with # of kernels scaling inversely with kernel width). They use a rigorous cross-validation procedure that involves separate steps for hyperparameter (λ) estimation and model selection, all of which I think is quite elegant. However, I’m wondering a bit about the ability of this modeling architecture to actually discriminate between different underlying tuning properties of each voxel (note that, of course, these approaches cannot speak to neural tuning properties, due to the intractable inverse problem relating neural and voxel-level signals). Let’s consider an extreme example: what if the authors had used 360 kernels, each with very narrow tuning width? Such a model should be able to fully approximate the wider-kernel models (each wide kernel can be accurately described as a weighted combination of the tiny-kernels). While there are of course concerns with overfitting, etc, in this toy example, I think it illustrates the trickiness associated with model selection in this way. As such, I think the authors need to demonstrate (via simulations) that this modeling approach can accurately and unambiguously uncover underlying ‘tuning’ properties of each voxel across a range of SNR and tuning scenarios (uni-modal voxel-level tuning; bimodal; random).

The reviewer suggests here to use simulations to demonstrate that our modeling framework uncovers the correct tuning across a range of different scenarios by running it on simulated voxel time courses with known tuning properties. We think this is a great way of demonstrating the strengths and limitations of our approach and thank the reviewer for suggesting it. In response, we set up various simulations as described in detail below.

First, we simulated voxel time courses with known tuning properties similar to how we built the design matrix for the actual modeling pipeline. To do so, we created the vHD-kernels reflecting the desired tuning profile and then computed the corresponding time course based on the actually observed virtual head direction (vHD) of a randomly chosen sample participant. This has several key advantages over simulating completely random time courses, because it captures the natural dependencies and correlations between different directions and how these are sampled over time. Note however that the choice of participant does not change the results. For each voxel we simulated all five scanning runs.

As suggested, we built simulated time courses for the assumed encoding of one random direction (unimodal model), two random directions (bimodal model) and randomly many random directions (random model) for each of the eight tuning width levels. The random numbers were sampled from a uniform distribution. Next, we added various degrees of Gaussian noise to these time courses ranging from one to ten standard deviations of the signal time course. For each of these noise levels and tuning widths, we simulated a total of

2500 voxels, which approximated a typical ROI-size in our data (See methods section). This resulted in 2500 unimodally tuned voxels for each of the 10 noise levels and each of the 8 different tuning widths for 5 simulated scanning runs; another 2500 bimodally tuned voxels and another 2500 randomly tuned voxels with equal noise, width levels and runs.

Next, we ran the full encoding model pipeline as described in the manuscript on these simulated time courses and assessed model performance after averaging across voxels. We predicted to see that the kernel width that served as the basis for the simulated time course would also lead to the best model performance when tested. The results for all model iterations, noise levels and tuning widths are depicted in a newly added Supplementary Fig. 4 as shown below.

These simulations confirmed the intuition of the reviewer that smaller kernels can serve as a good approximation of larger kernels. Kernels smaller than the true kernel overall led to higher model performance than kernels larger than the true kernel. This can be seen in Supplementary Fig. 4C showing higher values (on the Y-axis) for models above compared to below the true model. This seems true especially for low noise conditions. With increasing noise, the benefit of the smaller kernels vanishes, possibly related to the overfitting issue pointed out by the reviewer. Please note that increasing noise had an unsurprisingly detrimental effect on model performance. Because we are interested solely in the comparison between different model iterations within each noise level however, we normalized the effect of noise on model performance in Supplementary Fig. 4C.

Most importantly, despite the partial benefits of smaller over larger kernels, we want to strongly emphasize that the true kernel still won over both smaller and larger kernels in all cases, independent of noise level or tuning profile. Our modeling framework therefore uncovers the correct tuning width robustly. The true kernel width always led to the best model fit/prediction at least on ROI level. In the actual fMRI analyses, such averaging is done across voxels of an ROI, or by smoothing for the whole-brain group analyses (not the same as averaging but having a similar effect). We added a detailed description of these simulations including a new figure to the manuscript, hoping that these convince the reviewer that we present a robust modeling framework that uncovers the correct tuning properties of voxel activity assessed by fMRI. The following paragraphs were added to the manuscript.

Following changes were made to the manuscript (underlined):

Page 5

Notably, our results cannot be explained by biases in sampling (Supplementary Fig. 1), model regularization or data quality (Supplementary Fig. 3). Using simulated data, we further ensured that our analysis uncovers the true underlying tuning properties robustly across various noise levels and tuning profiles (unimodal, bimodal, random) (Supplementary Fig. 4).

The model uncovers directional tuning robustly in simulated time courses

To test the robustness of our model and to explore its limitations, we repeated the full model training and testing procedure for simulated voxel time courses with known tuning properties. As described in detail below, these simulations demonstrated that our model uncovers the correct kernel size robustly across various noise levels and tuning profiles (unimodal, bimodal and random directional tuning).

First, we simulated voxel time courses (Supplementary fig. 4A) with known tuning properties similar to how we built the design matrix for the actual modeling pipeline. To do so, we built the vHD-kernels reflecting the desired tuning profile and then computed the corresponding time course based on the actually observed vHD of a randomly chosen sample participant. This has several key advantages over simulating completely random time courses, because it captures the natural dependencies and correlations between different directions and how these are sampled over time. Note however that the choice of participant does not change these results presented here. For each voxel we simulated all five scanning runs.

We built simulated time courses for the assumed encoding of one random direction (unimodal model), two random directions (bimodal model) and randomly many random directions (random model) for each of the eight tuning width levels tested. The random numbers were sampled from a uniform distribution. Next, we added various degrees of Gaussian noise to these time courses ranging between one to ten standard deviations of the signal time course. For each of these noise levels and tuning widths, we simulated a total of 2500 voxels, approximating a typical ROI-size in our data (See methods section). This resulted in 2500 unimodally tuned voxels for each of the 10 noise levels and each of the 8 different tuning widths for 5 simulated scanning runs; another 2500 bimodally tuned voxels and another 2500 randomly tuned voxels with equal noise, width levels and runs.

Next, we ran the full encoding model pipeline as described in the manuscript on these simulated time courses and assessed model performance (i.e. the Pearson correlation between the observed and predicted (simulated) time course). We predicted to see that the kernel width that served as the basis for the simulated time course would also lead to the best model performance when tested. To test this, we averaged the model performance across voxels like in the actual fMRI-analysis.

We observed that, unsurprisingly, increasing noise had a detrimental effect on model performance (Supplementary Fig. 4B). However, because our framework solely relies on the comparison between different kernel sizes within each noise level, we normalized the effect of noise on model performance for visualization. Interestingly, kernels smaller than the true kernel tended to lead to better model performance than kernels larger than the true kernel (Supplementary Fig. 4C). This likely arises from the fact that larger kernels can be approximated by smaller, but not easily by even larger ones. With increasing noise, the benefit of the smaller kernels vanished, possibly related to stronger overfitting to the noise. Please note that in our imaging data, there was also no correlation between tSNR and model performance (Supplementary Fig. 3D).

Most importantly, and despite the partial benefits of smaller over larger kernels, the true kernel won over both smaller and larger kernels in all cases, independent of noise level or assumed tuning profile.

As predicted, the true kernel width therefore always resulted in the best model prediction, at least when averaged across voxels (Supplementary Fig. 4C). In our main analyses, this averaging is done across voxels of an ROI, or by smoothing for the whole-brain group analyses (not the same as averaging but having a similar effect). In sum, above-described simulations demonstrate that our behavioral encoding model framework uncovers the true underlying fMRI tuning robustly even in the present of noise and independent of the actual tuning profile (unimodal, bimodal or random).

Supplementary Fig. 4

Caption: Virtual head direction (vHD) simulations. A) We simulated voxel time courses with known vHD-tuning properties by combining the vHD of a randomly chosen participant with simulated vHD-tuning profiles (unimodal, bimodal, random directional tuning). Plotted here are time courses of a bimodally tuned voxel (60°) at three noise levels (0, 1 and 10 standard deviations of the time course). We modeled 2500 voxels with 5 scanning runs for each tuning profile, noise level and tuning width combination. B) Effect of noise. We depict the model performance over noise levels averaged across 2500 bimodally tuned voxels and tuning widths (blue line, SEM across voxels hidden behind line). The model performed better at lower noise levels. C) Simulation results for all tuning profiles, tuning widths and noise levels. The black outline marks which tuning width was expected to show the highest model performance. If a tuning width of e.g. 10° was simulated, the kernel width of 10° should have led to the best model performance. This is the case for all tuning profiles, noise levels and tuning widths tested. The effect of noise was normalized for visualization.

2. Model performance: related to the above, the authors are primarily comparing different versions of the same model architecture. But how does this type of model fare against, say, visual encoding models? Does this account for variance above and beyond other useful models, such as a motion energy model (Nishimoto et al, 2011)? And, related, can the authors demonstrate (again, via simulations), that there is no systematic bias associated with better model fits and wider/thinner tuning kernels? That is – under different ‘true’ voxel-level encoding models (unimodal thin vs thick tuning, bimodal tuning, random tuning), are there different expectations in terms of strength of model fit?

In the first part of the comment the reviewer asks how our encoding model compares to other visual encoding models such as e.g. one of motion energy. We believe this kind of model comparison combined with variance partitioning can indeed be very helpful to understand how different sensory or behavioral variables are encoded. In the present manuscript, we were interested in testing an encoding model of virtual head direction, a model that required many careful control analyses to be interpretable (which we provide in Supplementary Figure 1-9). In the course of the revisions presented here, we implemented and added several new analyses and data visualizations to the manuscript within a limited amount of time. We hope the reviewer understands that this imposes time constraints, requiring us to prioritize some analyses over others. A formal comparison of different encoding models was hence outside the scope of these revisions (and of the manuscript as a whole). However, we refer to this exciting possibility in the manuscript to guide future work into this direction as shown below.

Following changes were made to the manuscript (underlined):

Page 17

Looking forward, we see a wide range of future applications and exciting avenues to be explored. Future studies employing more dynamic model tests such as inverted encoding modeling (proof of principle in Supplementary Fig. 7) could address these questions directly by reconstructing the modeled behavioral features more dynamically (Gardner and Liu, 2019; Kriegeskorte and Douglas, 2019; Liu et al., 2018; Naselaris et al., 2011; Sprague et al., 2018, 2019). This could especially be helpful when combined with high temporal resolution measures such as magnetoencephalography (MEG) or intracranial recordings in combination with VR-tasks (Kunz et al., 2019) to monitor trial by trial changes in tuning as a function of behavioral performance. In addition, the influence of different sensory and behavioral variables on the present results could be further examined by removing the landmarks and testing which areas maintain their tuning, or by comparing the results to the ones of other encoding models such as one of motion energy (Nishimoto et al., 2011).

In the second part of the comment the reviewer asks about the relationship between tuning width and tuning strength and if we could use simulations to examine whether the different versions of the models are matched in predictive power. We believe this question is very related to the reviewer's

first question, and our response to it addresses central points of the question raised here. In short, our new simulations show that our modeling framework uncovers the correct tuning width robustly independent of noise level or tuning profile. This in turn also shows that the ‘true’ model led to the overall highest tuning strength. As outlined above, we observed a slight bias towards smaller kernels especially for low noise conditions, but in all cases tested, the true model still won (at least across voxels). On a related note, we believe that not only the simulations but also our empirical data speaks to this question. The fact that we do not see wider or more narrow kernels win every time, and that the tuning width even depends on memory performance in retrosplenial cortex even though the same models were tested in all cases, demonstrates that our results are not explained by biases in the tested models per se. For example, in Figure 3B we color-coded the tuning width of each voxel and revealed a width gradient along the parahippocampal cortex. If one tuning width was associated with overall better model fits, this tuning width should have always won in all voxels (given that there was no correlation between tSNR with tuning width and tuning strength (Supplementary Figure 3D)). As described in our response to comment 1, we added a section and a new Supplementary Figure 4 to the manuscript, which describe the simulation results in detail.

3. Effect of learning: can the authors make an effort to compare model fit properties for early and late halves of the experiment? I understand this makes the independent validation procedure tricky to implement, but even cross-validated model fitting here would be useful to bolster their argument that learning is more necessary for low-performance subjects, especially early in the experiment.

The reviewer here suggests comparing the model performance between different parts of the experiments and hence to elaborate more on potential effects of learning. This is a very interesting question indeed, and the suggested analysis could illuminate the relationship between the tuning and behavior further. In response to the reviewer’s comment (related to comment 1 by reviewer 2), we now added a full leave-one-run-out cross-validation procedure to the ROI-analysis, which we report in the new Supplementary Fig 8C. Each scanning run is now being tested individually using weights estimated in all other runs,

Before we discuss how the tuning strength changes over time, we would like to point out that we also computed the average tuning strength across all runs to see how robust the results reported in the original manuscript were. Critically, these analyses confirmed the key message of the original memory-split analysis (i.e. directional tuning reflects memory-error/performance). Even when cross-validated across runs, activity in higher order areas such as the entorhinal cortex strongly reflected behavioral performance, while activity in lower visual regions such as the early visual cortex did not. The effect does therefore not reflect biases or low-level visual differences between groups but arises in the higher levels of the cortical hierarchy in the parahippocampal cortex and the medial temporal lobe.

When comparing the tuning strength between runs and participant groups, we observed basically three types of tuning dynamics over time.

First, the tuning in some regions did not differ between participant groups. This was true for early visual and retrosplenial regions, like already reported in the original manuscript. In early visual cortex however the tuning tended to increase over time in both participant groups. Potentially, this could suggest that viewing behavior becomes more stereotypical as the environment becomes familiar, a hypothesis that could be specifically tested in future experiments using eye tracking. Please note that we point to future eye tracking experiments already in the manuscript. The tuning strength in the retrosplenial cortex did not show a clear increase or decrease over time.

Second, in intermediate stages of the cortical hierarchy such as the parahippocampal cortex, tuning strength strongly depended on memory performance, but did also not change over time. In fact, the tuning itself but also the difference in tuning between groups remained remarkably stable over time.

Third, in the hierarchically highest cortical regions we tested such as the hippocampus and the entorhinal cortex, the tuning depended on memory performance, but also showed systematic changes over time. Strikingly, while the tuning decreased in the low-error group over time, it even increased slightly in the high-error group. This is drastically different from the results of the EVC and RSC, but also from the ones in the parahippocampal cortex. One explanation could be that the participant groups follow distinct cognitive strategies, which get refined over time and with learning.

We thank the reviewer for suggesting these analyses and believe that they helped to emphasize one of the manuscript's key conclusions further. Directional tuning in medial temporal lobe regions reflects the behavioral performance, and this relationship might develop over time and experience.

Following changes were made to the manuscript (underlined):

Page 9

Contrary to our initial hypothesis, the differences between groups could reflect general differences in the cognitive strategy used or the participants' ongoing effort in encoding rather than retrieving a map of the environment. To investigate these effects post-hoc in more detail we examined how directional tuning strength developed in the course of the experiment. We performed leave-one-run-out cross-validation of our full modeling pipeline to obtain the directional tuning strength for each ROI not only for our original test run but for all runs. We found three main patterns of results (Supplementary Fig. 8C): EVC and RSC tuning strength again did not reflect memory performance, also not when cross-validated over runs. In contrast, PHG tuning strength did reflect memory performance robustly over runs. Strikingly, HPC and EC tuning also reflected memory performance, but the relationship between the two variables developed over time. Over scanning runs, tuning strength tended to increase in high-error participants and tended to decrease in low-error participants. Interestingly, on a behavioral level, we found that both groups approached the same level of memory performance in the course the experiment (Supplementary Fig. 2C-D), but that the low-memory-error

group had approached this performance level earlier in the experiment than the high-memory-error group (Supplementary Fig. 2D-E).

Page 12

“The direction of this effect and how it developed over time in MTL regions speaks to the idea that the two participant groups might follow different cognitive strategies and that the tuning reflects how well the environment has been encoded.”

Page 26

“To examine how directional tuning developed over time, we further performed leave-one-run-out cross-validation across all scanning runs (Supplementary Fig. 8C). We performed the ROI-level encoding modeling analysis (Fig. 5) for all runs, each time estimating the model weights on all runs but the one used for testing.”

Page 16

Hypothetically, because fMRI likely measures synaptic processing rather than output spiking (Logothetis et al., 2001), the tuning could reflect how much inputs a region receives. For example, the MTL of high-error participants might receive stronger inputs from perceptual regions, because these participants were still in the process of forming a cognitive map, which heavily relies on visual information. Accordingly, low-error participants might have formed a cognitive map already. Consistent with this idea, we found that both participant groups approached the same level of memory performance, but that participants with stronger tuning in higher-level regions showed steeper learning curves than those with weaker tuning (Supplementary Fig. 2C). This is also consistent with earlier reports showing that hippocampal activity tracks the amount of knowledge obtained at a given time rather than the accumulated absolute knowledge (Wolbers and Büchel, 2005) and with decreases in hippocampal activity in the course of spatial learning (Brodt et al., 2016).

New reference:

Brodt, S., Pöhlchen, D., Flanagin, V.L., Glasauer, S., Gais, S., and Schönauer, M. (2016). Rapid and independent memory formation in the parietal cortex. *Proc. Natl. Acad. Sci.* *113*, 13251–13256.

Page 16

Analyzing how the tuning developed over time revealed that such continuous learning-related disengagement could however explain only the results observed for the low-, not of the high-memory-error group. The latter even showed slight increases in directional tuning over time in the entorhinal cortex (Supplementary Fig. 8C). One account that would reconcile these findings is that the memory-dependent effects could reflect a difference in cognitive strategy used, which might develop and get refined in the course of spatial learning.

Supplementary Fig. 8C

Model performance across scanning runs

Caption: Model performance (see Fig. 5C) across scanning runs. We performed leave-one-run-out cross-validation for voxels within the ROIs to examine how directional tuning develops over time. In each cross-validation loop a different scanning run was taken as test run, while all others served as training runs. We plot the model performance (Z-score) for each run for high and low memory error participants (median-split by memory error) for each ROI in two formats: 1) mean (solid line) and standard error of the mean (shaded area) for each run, as well as 2) the average model performance across runs as group-level whisker-boxplots.

Following statement on page 16 was removed:

“In the present study, we did not investigate how directional representations develop over time.”

Minor:

1. Can the authors make some effort to describe the types of tuning they observe at the voxel level? Do voxels prefer individual directions? Multiple directions? Are some directions over-represented in the neural code compared to others?

The reviewer raises an intriguing question about the underlying tuning profiles. Related to question 2 by Reviewer 2, we added a new figure to the manuscript addressing this question directly (see below). We depict the model weights across directions 1) for a typical voxel of each region, 2) the across-voxel-average within these regions and c) the across-voxel-average after aligning the peak-direction across voxels. The latter was done because the preferred directions could differ across voxels but still follow a similar tuning principle (e.g. bimodality), which should show in this plot. Interestingly, we did not observe a clear trend towards any tuning profile that was consistent across voxels, specific to a certain region or that differed across regions. This suggests that a given voxel does not represent all directions equally (potentially providing the basis for the observable fMRI signal), and that the represented directions did not cluster within each voxel (in a for us conceivable way).

Supplementary fig. 5

A) Model weights across directions for exemplary voxels & ROIs

Caption: Tuning profiles for randomly selected sample voxels and regions of interest (ROIs). We plot the model weights across directions for one exemplary voxel of each ROI (left panel) and the model weights averaged across voxels of these ROIs (middle panel). To test whether there are tuning profiles that were consistent over voxels (e.g. uni-, bi-, trimodality) but averaged out across voxels, we additionally aligned the peak model weight across voxels (right panel). We plot the mean (solid line) and one standard deviation (shaded area) across voxels. ROIs: early visual cortex (EVC), retrosplenial cortex (RSC), parahippocampal gyrus (PHG), hippocampus (HPC) and posteromedial entorhinal cortex (pmEC). This figure shows that different voxels have distinct tuning profiles and prefer different directions.

2. Is it possible to include an example figure showing predicted and actual time-courses for an example voxel? (similar to the visual population receptive field papers; e.g., Dumoulin & Wandell, 2008)

Absolutely, we fully share the reviewer's appreciation for this type of figure and added one to our new Supplementary figure 5B. Please note that model fits are often visualized on the data the model weights were estimated on (e.g. also in case of Dumoulin & Wandell 2008). Because our framework heavily relies on cross-validation, we however believe it is more useful to compare predicted and real time courses in our independent test data. These might look less impressive than other visualizations mentioned above, but we hope the reviewer agrees that this is more informative than plotting the training model fit here.

Supplementary Fig. 5B

B) Example of model fit to independent test data

Caption: Example model fit to independent test data. We depict the predicted time course (black) of two sample voxels from the early visual cortex (EVC) superimposed onto the actually observed time course of these voxels in the test run. The model weights used to build the predicted time course were estimated on independent training data. All time courses were z-scored.

3. The authors should describe the task in more detail – did the initial learning component occur inside the scanner? How is the fixation period modeled in the fMRI analyses? Where were the objects placed in the environment? What were the objects?

The initial learning session that taught the participants' how to do the task was done outside the scanner and they were allowed to ask questions. In response to the reviewer's question, we have extended the task description in the "Virtual reality task" section. Additionally, we added two new figures of the object locations along with the objects themselves to Supplementary figure 2 as shown below.

Page 20

"The location of these objects differed across participants (Supplementary Fig. 2A) and the six objects were randomly drawn from a set of overall twelve objects (Supplementary Fig. 2B) leading to unique location-object associations for each participant."

Page 20

„In order to explain the task and to familiarize participants with it, they performed a similar task on a desktop computer setup with different objects in a different virtual environment prior to scanning."

Supplementary figure 2AB

[Redacted]

Caption: Supplementary figure 2: Memory performance during the object-location memory task. We plot object locations (A, white dots) for all objects (B) and participants. Each participant was tested using a six of these objects and locations. [...]

The fixation period is not modeled but is a) independent of virtual head direction and orthogonal to the effects of interest and b) cannot explain the effects reported here due to the strict cross-validation procedure we adhere.

4. The variance explained seems to be extremely low (Supp Fig. 5 – Pearson's $r \sim 0.1$) – given these low values, how much could we expect to differentiate different versions of these models?

Yes, the variance explained is low indeed. Especially in the vision science domain however encoding models have reached higher prediction accuracies than this (under more controlled experimental conditions), demonstrating that the approach as a whole is well

suited to analyze fMRI data (e.g. see Naselaris et al. 2015). Here, we use this approach to study navigation-related variables under naturalistic (hence minimally constrained & noisy) conditions, leading to model accuracies that are low, but still well within the range of other reported effects in the spatial navigation literature. Naturally, this also means that the difference across models is not large. However, we would like to point out three aspects to consider. First, as pointed out by the reviewer, the differences across models might not be expected to be large because the various versions of the models also resemble each other to some degree. Second, because the effects are small, we rely on group-level inferences to only consider effects present in a majority of participants. Finally, and overlapping with the reviewer's first comment, we now validated the model on simulated data, showing that it uncovers the true underlying tuning even when all versions of the model compete against each other under high-noise conditions. We therefore believe that our approach and findings are valuable to the neuroimaging community despite the explained variance being low, and might serve as a 'stepping stone' for the development of even better behavioral encoding models in the future.

5. The authors mention a field map was acquired during imaging, but it's not clear how/if this scan was used during data (pre)processing.

We thank the reviewer for pointing us to this missing information regarding preprocessing. The fieldmap was used to unwarped the images and we added a statement to the "Preprocessing" section accordingly.

Page 22

"...functional images were realigned and unwarped using SPM8..."

6. On pg 22 the authors describe a procedure where they exclude voxels that are not well-fit by the regularized encoding model (where a good lambda could not be identified) – how many voxels were excluded?

Across all model iterations (tuning widths) and participants, this procedure excluded around 15.5% of voxels from determining the regularization parameter lambda. Please note that after this exclusion, lambda was averaged across the remaining voxels, and then used to estimate the final model weights serving for later model test. The above-described voxel exclusion was done only for determining lambda, not for the full training-test procedure. We clarified this point in the "Model training" section as follows:

Page 24

"Overall, this excluded ~15.5% of voxels (median across participants and model iterations), however only from determining the final λ , not from the full training-test procedure in which the final λ was applied to all voxels."

Reviewers' Comments:

Reviewer #1:

Remarks to the Author:

The authors have made a direct and good faith effort to respond to my comments. The manuscript is clearer and ready for publication.

Reviewer #2:

Remarks to the Author:

The revised manuscript is greatly improved. The authors have addressed all my concerns and I have no further issues to raise. This is a beautiful study, rigorously conducted, and a very interesting contribution to the literature.

Reviewer #3:

Remarks to the Author:

I appreciate the authors' exceptionally thorough consideration of my comments (and those of the other reviewers), and think the resulting revised manuscript is substantially stronger as a result. I especially love the inclusion of the simulations – that's such a nice result, extremely well done! I also think it's fantastic that the authors are including detailed source data files for each figure, as well as the simulation/analysis code and example data. They set a wonderful example for others to follow! I have no further reservations about endorsing this manuscript for publication in Nature Communications, and wish to congratulate the authors on a job beautifully done!

Point-by-point response:

“Behavior-dependent directional tuning in the human visual-navigation network”

Thank you for your helpful comments regarding our manuscript. We are delighted and grateful to have received such positive feedback by you and the reviewers. Please find our responses to the individual comments below.

Reviewer comments:

Reviewer #1:

The authors have made a direct and good faith effort to respond to my comments. The manuscript is clearer and ready for publication.

Reviewer #2:

The revised manuscript is greatly improved. The authors have addressed all my concerns and I have no further issues to raise. This is a beautiful study, rigorously conducted, and a very interesting contribution to the literature.

Reviewer #3:

I appreciate the authors' exceptionally thorough consideration of my comments (and those of the other reviewers), and think the resulting revised manuscript is substantially stronger as a result. I especially love the inclusion of the simulations – that's such a nice result, extremely well done! I also think it's fantastic that the authors are including detailed source data files for each figure, as well as the simulation/analysis code and example data. They set a wonderful example for others to follow! I have no further reservations about endorsing this manuscript for publication in Nature Communications, and wish to congratulate the authors on a job beautifully done!

We are extremely grateful for the insightful comments by the reviewers during the first round of revisions and are delighted to receive such positive feedback now. We thank all reviewers for their effort and time and for helping tremendously to improve the quality of our manuscript.